# OMNIRE: OMNI URBAN SCENE RECONSTRUCTION

**Ziyu Chen**[1,6*]  **Jiawei Yang**[6]  **Jiahui Huang**[5]  **Riccardo de Lutio**[5]
**Janick Martinez Esturo**[5]  **Boris Ivanovic**[5]  **Or Litany**[2,5]  **Zan Gojcic**[5]
**Sanja Fidler**[3,5]  **Marco Pavone**[4,5]  **Li Song**[1]  **Yue Wang**[5,6]

1Shanghai Jiao Tong University  [2]Technion  [3]University of Toronto
[4]Stanford University  [5]NVIDIA Research  [6]University of Southern California

## ABSTRACT

We introduce `OmniRe`, a comprehensive system for efficiently creating high-fidelity digital twins of dynamic real-world scenes from on-device logs. Recent methods using neural fields or Gaussian Splatting primarily focus on vehicles, hindering a holistic framework for all dynamic foregrounds demanded by downstream applications, e.g., the simulation of human behavior. `OmniRe` extends beyond vehicle modeling to enable accurate, full-length reconstruction of diverse dynamic objects in urban scenes. Our approach builds scene graphs on 3DGS and constructs multiple Gaussian representations in canonical spaces that model various dynamic actors, including vehicles, pedestrians, cyclists, and others. `OmniRe` allows holistically reconstructing any dynamic object in the scene, enabling advanced simulations (~60 Hz) that include human-participated scenarios, such as pedestrian behavior simulation and human-vehicle interaction. This comprehensive simulation capability is unmatched by existing methods. Extensive evaluations on the Waymo dataset show that our approach outperforms prior state-of-the-art methods quantitatively and qualitatively by a large margin. We further extend our results to 5 additional popular driving datasets to demonstrate its generalizability on common urban scenes. Code and results are available at omnire.

## 1 INTRODUCTION

Creating photorealistic digital twins of 4D real-world is valuable for enabling high-fidelity simulation, robust algorithm training and evaluation. As autonomous driving algorithms increasingly adopt end-to-end models, the need for scalable and domain-gap-free simulation environments, where these systems can be evaluated in closed-loop, is becoming more evident. While traditional artist-generated assets are reaching their limits in scale, diversity, and realism, recent advances in data-driven methods offer a strong alternative by creating realistic digital twins directly from real-world sensor data. Indeed, neural radiance fields (NeRFs) (Mildenhall et al., 2020; Barron et al., 2021; Yang et al., 2023b; Guo et al., 2023; Yang et al., 2023a; Wu et al., 2023b) and Gaussian Splatting (GS) (Kerbl et al., 2023; Yan et al., 2024) have emerged as powerful tools for reconstructing 3D scenes with high levels of visual and geometric fidelity. However, accurately and holistically reconstructing dynamic urban scenes remains a significant challenge, especially due to the diverse dynamic actors and their complex rigid and non-rigid motions in real-world environments.

Several works have already tried to tackle this challenge. Early methods typically ignore dynamic actors and reconstruct only static parts of the scene (Tancik et al., 2022; Martin-Brualla et al., 2021; Rematas et al., 2022; Guo et al., 2023). Subsequent works aim to reconstruct the dynamic scenes by either **(i)** modeling the scenes as a combination of a static and time-dependent dynamic neural field, where the static-dynamic decomposition is an emergent property (Yang et al., 2023a; Turki et al., 2023), or **(ii)** building a scene graph, in which dynamic actors and the static background are represented as nodes and reconstructed in their canonical frame. The nodes of the scene graph are connected with edges that encode relative transformation representing the motion of each actor through time (Ost et al., 2021; Kundu et al., 2022; Yang et al., 2023b; Wu et al., 2023b; Tonderski

---

*Work done during a research internship at University of Southern California.
✉ Ziyu Chen <ziyu.sjtu@gmail.com>  Yue Wang <yue.w@usc.edu>

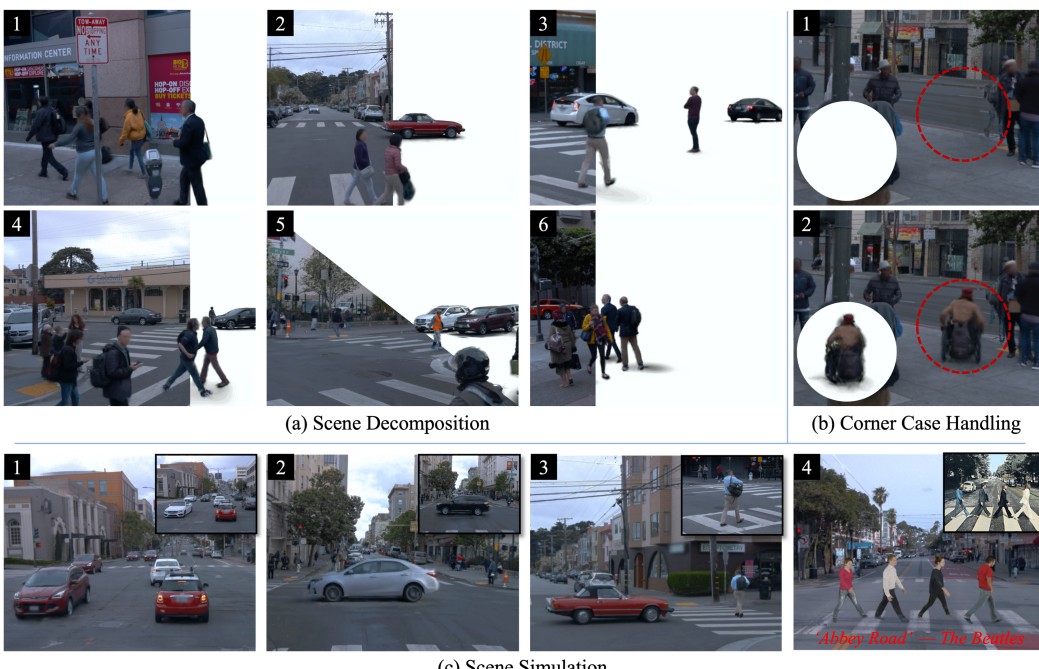

Figure 1: (a) Decomposition of different parts of a scene. (b) Out-of-distribution categories that are overlooked by previous methods can be accurately handled by `OmniRe`. (c) `OmniRe` enables diverse applications including vehicle editing (c1, c2), human-vehicle interaction (c3), human behavior simulation (c4), etc.

et al., 2024; Fischer et al., 2024b). But both approaches fall short of meeting the requirements for comprehensive and interactive digital twins: while providing a more general formulation, methods of **(i)** lack editability and cannot be directly controlled with classical behavior models. Previous methods following **(ii)** still focus primarily on vehicles, which can be represented as rigid bodies, thereby largely neglecting other vulnerable road users (VRUs) such as pedestrians and cyclists that are fundamental and critical in urban scene simulation.

To fill this critical gap, our work aims to model all dynamic actors , including vehicles, pedestrians, and cyclists, and many others, in a manner that allows for interactive simulation. This leads to two primary challenges: **(i)** developing a unified approach for modeling diverse non-rigid dynamic actors, given the wide range of non-rigid categories in real-world scenes; **(ii)** giving specific focus on humans, as their behavior is critical for decision-making in scenarios like driving, where pedestrian actions directly impact safety. Thus, precise joint-level reconstruction (Lei et al., 2023; Jiang et al., 2022; Kocabas et al., 2024) is crucial for fine control of human behavior in the simulator. To address the specific challenge of modeling human actors, we must consider several additional complexities. First, in-the-wild scenarios present significant challenges for capturing human motion dynamics due to unfavorable sensor observations and cluttered environments with frequent occlusions (Wang et al., 2024; Yang et al., 2021; Wang et al., 2023). Furthermore, reconstructing high-fidelity human appearance from sparse sensor data beyond mere geometry dynamics adds additional complexity. Lastly, interactions with large equipment, such as wheelchairs or strollers, which cannot be represented by explicit templates (e.g., SMPL), further complicate both appearance and geometry modeling.

To address these challenges, we propose an "omni" system capable of handling diverse actors for urban digital twins. Our method `OmniRe` efficiently reconstructs high-fidelity urban scenes that include static backgrounds, driving vehicles, and non-rigidly moving dynamic actors (see Fig. 1). Specifically, we construct a dynamic neural scene graph (Ost et al., 2021) based on 3D Gaussian Splatting (Kerbl et al., 2023), with dedicated Gaussian representations for different kinds of dynamic actors in their local canonical spaces. In our framework, backgrounds and vehicles are represented as static Gaussians, while vehicles undergo rigid body transformations to simulate their motion over time. For non-rigid actors, we incorporate the SMPL model to enable joint-level control for pedestrians using dynamic Gaussians, as SMPL provides a prior template geometry for 3DGS initialization and explicit control for modeling desired human behaviors, which is advantageous for downstream simulation applications. To extract SMPL parameters for human motion modeling, we

designed a novel human body pose estimation pipeline dedicated to driving logs with multi-camera setups and severe in-the-wild occlusions. For other template-less dynamic actors, we propose a shared deformation field approach in a self-supervised manner. This framework enables a unified representation of all non-rigid categories and achieves specialized joint-level control for pedestrians. Thus, OmniRe allows for accurate representation and controllable reconstruction of most objects of interest in the scene. Notably, our representation is directly amenable to behavior and animation models that are commonly used in AV simulation (*e.g.*, Fig. 1-(c)).

To summarize, we make the following contributions:

- We introduce OmniRe, a holistic framework for dynamic urban scene reconstruction that embodies the "omni" principle of dynamic category coverage and representation flexibility. OmniRe leverages dynamic neural scene graphs based on Gaussian representations to unify the reconstruction of static backgrounds, driving vehicles, and non-rigidly moving dynamic actors (§ 4). It enables high-fidelity scene reconstruction and human-centered simulations, such as pedestrian behavior and human-vehicle interaction—capabilities unmatched by existing methods. (§ 5).
- We address the challenges of modeling humans and other dynamic actors from driving logs such as occlusion, cluttered environments, and the limitations of existing human pose prediction models (§ 4.2). We demonstrate our method on six popular driving datasets to show its generalizability in urban driving scenes (project page). While our findings are based on AV scenarios, they can generalize to other domains.
- We perform extensive experiments and ablations to demonstrate the benefits of our holistic framework. OmniRe achieves state-of-the-art performance in scene reconstruction and novel view synthesis (NVS), significantly outperforming previous methods in terms of full image metrics (+1.88 PSNR for reconstruction and +2.38 PNSR for NVS). The differences are pronounced for dynamic actors, such as vehicles (+1.18 PSNR), and humans (+4.09 PSNR for reconstruction and +3.06 PSNR for NVS) (Tab. 1).

## 2 RELATED WORK

**Dynamic Scene Modeling.** Neural representations are dominating novel view synthesis (Mildenhall et al., 2020; Barron et al., 2022; 2021; Müller et al., 2022; Fridovich-Keil et al., 2022; Kerbl et al., 2023). These have been extended in different ways to enable dynamic scene reconstruction. *Deformation-based* approaches (Pumarola et al., 2020; Park et al., 2021a; Tretschk et al., 2021; Park et al., 2021b; Cai et al., 2022) and recently DeformableGS (Yang et al., 2023c) and (Wu et al., 2023a) propose to model dynamic scenes using a 3D neural representation for the canonical space, coupled with a deformation network mapping time-dependent observations to canonical deformations. These are generally limited to small scenes with limited movement, making them inadequate for challenging urban dynamic scenes. *Modulation-based* techniques operate by directly feeding the image timestamps (or latent codes) as an additional input to a neural representation (Xian et al., 2021; Li et al., 2021; 2022; Luiten et al., 2024). However, this generally results in an underconstrained formulation, therefore requiring additional supervision, such as depth (Li et al., 2021) and optical flow (Xian et al., 2021), or multi-view inputs captured from synchronized cameras (Li et al., 2022; Luiten et al., 2024). $D^2$NeRF (Wu et al., 2022) proposed to expand on this formulation by partitioning the scene into static and dynamic fields. Following this, SUDS (Turki et al., 2023) and EmerNeRF (Yang et al., 2023a) have shown impressive reconstruction ability for dynamic autonomous driving scenes. However, they model all dynamic elements using a single dynamic field, rather than modeling each separately, thus they lack controllability, limiting their practicality as sensor simulators. *Explicit decomposition* of the scene into separate agents enables controlling them individually. These agents can be represented as bounding boxes in a scene graph as in Neural Scene Graphs (NSG) (Ost et al., 2021) that is widely adopted in UniSim (Yang et al., 2023b), MARS (Wu et al., 2023b), NeuRAD (Tonderski et al., 2024), ML-NSG (Fischer et al., 2024b) and recent Gaussian-based works StreetGaussians (Yan et al., 2024), DrivingGaussians (Zhou et al., 2023), and HUGS (Zhou et al., 2024). However, these approaches handle only rigid objects due to limitations of time-independent representations (Ost et al., 2021; Wu et al., 2023b; Yang et al., 2023b; Zhou et al., 2023; 2024; Yan et al., 2024; Tonderski et al., 2024; Fischer et al., 2024b) or limitations of deformation-based techniques (Yang et al., 2023c; Huang et al., 2023). A recent concurrent work Fischer et al. (2024a) also considers non-rigid modeling using a deformation field, addressing a subset of the challenges in modeling holistic dynamics, but does not address fine-grained human models that allow flexible control. To address them, OmniRe proposes a

Gaussian scene graph that incorporates various Gaussian representations for both rigid and non-rigid objects, providing extra flexibility and controllability for diverse actors.

**Human Modeling.** Human bodies have variable appearance and complex motions, calling for dedicated modeling techniques. NeuMan (Jiang et al., 2022) proposes to employ the SMPL body model (Loper et al., 2015) to warp ray points to a canonical space. This approach enables the reconstruction of non-rigid human bodies and warrants fine control. Similarly, recent works such as GART (Lei et al., 2023), GauHuman (Hu & Liu, 2023) and HumanGaussians (Kocabas et al., 2024) have combined the Gaussian representation and the SMPL model. However, these methods are not directly applicable in-the-wild. As for recovering human dynamics in driving scenes, Yang et al. (2021) focuses on shape and pose reconstruction for LiDAR simulation, while Wang et al. (2023; 2024) aim to recreate natural and accurate human motion from partial observations. However, these methods focus solely on shape and pose estimation and are limited in appearance modeling. In contrast, our method not only models human appearance but also integrates this modeling within a holistic scene framework, to achieve a comprehensive solution. Urban scenes typically involve numerous pedestrians, with sparse observation, often accompanied by severe occlusion. We analyze these challenges in detail and address them in § 4.2.

## 3 PRELIMINARIES

**3D Gaussian Splatting.** First introduced in Kerbl et al. (2023), 3D Gaussian Splatting (3DGS) represents scenes via a set of colored blobs $\mathcal{G} = \{g\}$ whose intensity distribution is a Gaussian. Each Gaussian (blob) $g = (o, \boldsymbol{\mu}, \mathbf{q}, s, c)$ contains the following attributes: opacity $o \in (0, 1)$, mean position $\boldsymbol{\mu} \in \mathbb{R}^3$, rotation $\mathbf{q} \in \mathbb{R}^4$ represented as a quaternion, anisotropic scaling factors $s \in \mathbb{R}_+^3$, and view-dependent colors $c \in \mathbb{R}^F$ represented as spherical harmonics (SH) coefficients. To compute the color $C$ of a pixel, Gaussians overlapping with this pixel are sorted by their distance to the camera center (sorted by $i \in \mathcal{N}$) and $\alpha$-blended: $C = \sum_{i \in \mathcal{N}} c_i \alpha_i \prod_{j=1}^{i-1} (1 - \alpha_j)$, where $\alpha_i$ is computed as $\alpha_i = o_i \exp(-\frac{1}{2}(\mathbf{p} - \boldsymbol{\mu}_i)^T \boldsymbol{\Sigma}_i^{-1} (\mathbf{p} - \boldsymbol{\mu}_i))$, $\boldsymbol{\Sigma}_i$ is the 2D projection covariance. We further define the application of a rigid (affine) transformation $\mathbf{T} = (\mathbf{R}, \mathbf{t}) \in \mathbb{SE}(3)$ to all Gaussians in the set as: $\mathbf{T} \otimes \mathcal{G} = (o, \mathbf{R}\boldsymbol{\mu} + \mathbf{t}, \text{Rot}(\mathbf{R}, \mathbf{q}), s, c)$, where $\text{Rot}(\cdot)$ denotes rotating the quaternion by the rotation matrix.

**Skinned Multi-Person Linear (SMPL) Model.** SMPL (Loper et al., 2015) is a parametric human body model that combines the advantages of a parametric mesh with linear blending skinning (LBS) to manipulate body shape and pose. At its core, SMPL uses a template mesh $\mathcal{M}_h = (\mathcal{V}, \mathcal{F})$ defined in a canonical rest pose, parameterized by $n_v$ vertices $\mathcal{V} \in \mathbb{R}^{n_v \times 3}$. The template mesh can be shaped and transformed using shape parameters $\boldsymbol{\beta}$ and pose parameters $\boldsymbol{\theta}$: $\mathcal{V}_S = \mathcal{V} + B_S(\boldsymbol{\beta}) + B_P(\boldsymbol{\theta})$, where $B_S(\boldsymbol{\beta}) \in \mathbb{R}^{n_v \times 3}$ and $B_P(\boldsymbol{\theta}) \in \mathbb{R}^{n_v \times 3}$ are the $xyz$ offsets to individual vertices (Kocabas et al., 2024) and $\mathcal{V}_S$ are the vertex locations in the shaped space.

To further deform the vertices $\mathcal{V}_S$ to achieve the desired pose $\boldsymbol{\theta}'$, SMPL utilizes pre-defined LBS weights $\boldsymbol{W} \in \mathbb{R}^{n_k \times n_v}$ and the joint transformations $\boldsymbol{G}$ to define the deformation of each vertex $\boldsymbol{v}_i$: $\boldsymbol{v}_i' = (\sum_k \boldsymbol{W}_{k,i} \boldsymbol{G}_k) \boldsymbol{v}_i$, where $n_k$ is the number of joints, and the joint transformations $\boldsymbol{G}$ are derived from the source pose $\boldsymbol{\theta}$, the target pose $\boldsymbol{\theta}'$ and shape $\boldsymbol{\beta}$. The pose parameters include the body pose component $\boldsymbol{\theta}_b \in \mathbb{R}^{23 \times 3 \times 3}$ and the global orientation component $\boldsymbol{\theta}_g \in \mathbb{R}^{3 \times 3}$. For more details of SMPL, we refer readers to Loper et al. (2015). Our method obtains pose parameters $\boldsymbol{\theta}$ for each pedestrian across all frames, as well as their individual shape parameters $\boldsymbol{\beta} \in \mathbb{R}^{10}$, these pose sequences initialize the non-rigid dynamics of pedestrians. The detailed process is described in § 4.2.

## 4 METHOD

As overviewed in Fig. 2, we build a comprehensive 3DGS framework that holistically reconstructs both the static background and diverse *movable* entities. We discuss our systematic approach that represents different semantic classes with diverse Gaussian representations in § 4.1, highlighting that this complex yet efficient system-level framework is one of our primary contributions. Modeling humans in unconstrained environments is particularly challenging due to the complexity of human motions and the difficulty of accurately modeling geometry and appearance due to severe occlusions in the wild. We present our approach to this problem in § 4.2, which significantly expands our

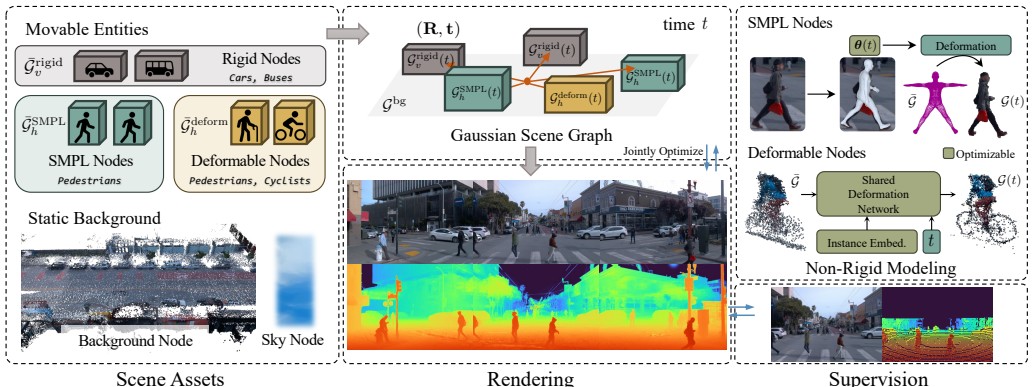

Figure 2: **Method Overview.** Gaussians of all foreground models are defined in their local or canonical spaces. At a given time $t$, the Gaussians are deformed and transformed into the world space, forming a Gaussian scene graph together with background Gaussians to model the entire scene. The Gaussians in the scene graph are rasterized to render images and depth, and are jointly optimized using reconstruction losses. We utilize SMPL Gaussians to model non-rigid human bodies and deformable Gaussians to handle out-of-distribution non-rigid categories.

effectiveness in common driving scenes. Lastly, we show how the scene representation is end-to-end optimized to obtain faithful and controllable reconstructions in § 4.3.

## 4.1 DYNAMIC GAUSSIAN SCENE GRAPH MODELING

**Gaussian Scene Graph.** To allow for flexible control of diverse *movable* objects in the scene without sacrificing reconstruction quality, we opt for a *Gaussian Scene Graph* representation. Our scene graph is composed of the following nodes: (1) a *Sky Node* representing the sky that is far away from the ego-car, (2) a *Background Node* representing the static scene background such as buildings, roads, and vegetation, (3) a set of *Rigid Nodes*, each representing a rigidly movable object such as a vehicle, (4) a set of *Non-rigid Nodes* that model non-rigid individuals, e.g. pedestrians and cyclists. Nodes of type (2,3,4) can be converted directly into world-space Gaussians which we will introduce next. These Gaussians are concatenated and rendered using the rasterizer proposed in Kerbl et al. (2023). The Sky Node is represented by an optimizable environment texture map, similar to Chen et al. (2023), rendered separately, and composited with the rasterized Gaussian image with simple alpha blending.

**Background Node.** The background node is represented by a set of static Gaussians $\mathcal{G}^{\mathrm{bg}}$. These Gaussians are initialized by accumulating the LiDAR points and additional points generated randomly in accordance with the strategy described in Chen et al. (2023).

**Rigid Nodes.** Gaussians representing the vehicles (*e.g.* cars or trucks) are defined as $\bar{\mathcal{G}}_v^{\mathrm{rigid}}$ in the object's local space (denoted by the upper bar), where $v$ is the index of the vehicle/node. While the Gaussians within a vehicle will not change over time in the local space, the positions of Gaussians in world space will change according to the vehicle's pose $\mathbf{T}_v \in \mathbb{SE}(3)$. At a given time $t \in \mathbb{R}$, the Gaussians are transformed into world space by simply applying the pose transformation:

$$\mathcal{G}_v^{\mathrm{rigid}}(t) = \mathbf{T}_v(t) \otimes \bar{\mathcal{G}}_v^{\mathrm{rigid}}. \tag{1}$$

**Non-Rigid Nodes.** Non-rigid individuals are often overlooked by previous methods (Zhou et al., 2024; Yan et al., 2024; Zhou et al., 2023; Fischer et al., 2024b) due to the modeling complexity, despite their importance for human-centered simulation. Unlike rigid vehicles, non-rigid dynamic classes such as pedestrians and cyclists, require extra consideration of both their global movements in world space and their continuous deformations in local space to accurately reconstruct their dynamics. To enable a reconstruction that fully explains the underlying geometry, we further subdivide the non-rigid nodes into two categories: *SMPL Nodes* for walking or running pedestrians with SMPL templates that enable joint-level control and *Deformable Nodes* for out-of-distribution non-rigid instances (such as cyclists and other template-less dynamic entities).

**Non-Rigid SMPL Nodes.** As introduced in § 3, SMPL provides a parametric way of representing human poses and deformations, and we hence use the model parameters $(\boldsymbol{\theta}(t), \boldsymbol{\beta})$ to drive the 3D

Gaussians within the nodes. Here $\boldsymbol{\theta}(t) \in \mathbb{R}^{24 \times 3 \times 3}$ represents the human posture that changes over time $t$. For each node indexed by $h$, We tessellate the SMPL template mesh $\mathcal{M}_h$ instantiated from the resting pose (the '*Da*' pose) with 3D Gaussians $\bar{\mathcal{G}}_h^{\text{SMPL}}$ using a strategy similar to Lei et al. (2023), where each Gaussian is binded to its corresponding vertex of $\mathcal{M}_h$. The world-space Gaussians for each node can be then computed as:

$$\mathcal{G}_h^{\text{SMPL}}(t) = \mathbf{T}_h(t) \otimes \text{LBS}(\boldsymbol{\theta}(t), \bar{\mathcal{G}}_h^{\text{SMPL}}). \tag{2}$$

Here $\mathbf{T}_h(t) \in \mathbb{SE}(3)$ is the global pose of the node at time $t$, and $\text{LBS}(\cdot)$ is the linear blend skinning operation that deforms the Gaussians according to the SMPL pose parameters. In order to compute the LBS operator, one first precomputes the skinning weights of each Gaussian in $\bar{\mathcal{G}}_h^{\text{SMPL}}$ w.r.t. the SMPL key joints. Once $\boldsymbol{\theta}$ changes over time, the key joints' transformations are updated and linearly interpolated onto the Gaussians to obtain the final deformed positions and rotations, while other attributes in the Gaussian remain unchanged. Crucially, it is highly challenging to accurately optimize the SMPL poses $\boldsymbol{\theta}(t)$ from scratch just with sensor observations, even for single-person or indoor scenarios (Jiang et al., 2022; Lei et al., 2023; Kocabas et al., 2024). Hence a rough initialization of $\boldsymbol{\theta}(t)$ is typically needed, whose details are deferred to a dedicated section § 4.2.

**Non-Rigid Deformable Nodes.** These nodes act as a unified representation for other significant non-rigid instances, including those that fall beyond the scope of SMPL modeling, such as extremely faraway pedestrians for which even state-of-the-art SMPL estimators cannot provide accurate estimations, or out-of-distribution, template-less non-rigid individuals. Hence, we propose to use a general deformation network $\mathcal{F}_\varphi$ with parameter $\varphi$ to fit the non-rigid motions within the nodes. Specifically, for node $h$, the world-space Gaussians are defined as:

$$\mathcal{G}_h^{\text{deform}}(t) = \mathbf{T}_h(t) \otimes \left( \bar{\mathcal{G}}_h^{\text{deform}} \oplus \mathcal{F}_\varphi(\bar{\mathcal{G}}_h^{\text{deform}}, \boldsymbol{e}_h, t) \right), \tag{3}$$

where the deformation network generates the changes of the Gaussian attributes from time $t$ to the canonical space Gaussians $\bar{\mathcal{G}}_h^{\text{deform}}$, outputting the changes in position $\delta\boldsymbol{\mu}_h(t)$, rotation $\delta\mathbf{q}_h(t)$, and the scaling factors $\delta\boldsymbol{s}_h(t)$. The changes are applied back to $\bar{\mathcal{G}}_h^{\text{deform}}$ with the $\oplus$ operator that internally performs a simple arithmetic addition that results in $(o, \boldsymbol{\mu} + \delta\boldsymbol{\mu}(t), \mathbf{q} + \delta\mathbf{q}(t), \boldsymbol{s} + \delta\boldsymbol{s}(t), \boldsymbol{c})$. Notably, previous approaches such as Yang et al. (2023c) utilizes a single deformation network for the entire scene, and usually fail in highly complex outdoor dynamic scenes with rapid movements. On the contrary, in our work, we define a per-node deformation field which has much more representation power. To maintain computational efficiency, the network weights $\varphi$ are shared and the identities of the nodes are disambiguated via an instance embedding parameter $\boldsymbol{e}_h$. Experimental results in § 5.2 show that deformable Gaussians are essential for achieving good reconstruction quality.

**Sky Node.** We use a separate optimizable environmental map to fit the sky color from viewing directions. Compositing the sky image $C_{\text{sky}}$ with the rendered Gaussians $C_\mathcal{G}$ consisting of $(\mathcal{G}^{\text{bg}}, \{\mathcal{G}_v^{\text{rigid}}\}, \{\mathcal{G}_h^{\text{SMPL}}\}, \{\mathcal{G}_h^{\text{deform}}\})$, we obtain the final rendering as:

$$C = C_\mathcal{G} + (1 - O_\mathcal{G})C_{\text{sky}}, \tag{4}$$

where $O_\mathcal{G} = \sum_{i=1}^{N} \alpha_i \prod_{j=1}^{i-1}(1 - \alpha_j)$ is the rendered opacity mask of Gaussians.

## 4.2 RECONSTRUCTING IN-THE-WILD HUMANS

Reconstructing humans from driving logs faces challenges as in-the-wild pose estimators (Goel et al., 2023; Rajasegaran et al., 2022) are typically designed for single video input and often miss predictions in occlusion cases. We designed a pipeline that addresses these limitations to predict accurate and temporally consistent human poses from multi-view videos with frequent occlusions.

Formally, given a set of 3D tracklets for $N$ pedestrians $\mathbf{T} = \{\mathbf{T}_h\}_{h=0}^{N-1}$ from the dataset, our goal is to obtain the corresponding SMPL pose sets: $\boldsymbol{\theta} = \{\boldsymbol{\theta}_h\}_{h=0}^{N-1}$. Here, $\mathbf{T}_h$ and $\boldsymbol{\theta}_h$ (For brevity, $(t)$ is omitted) represent the boxes sequence and body pose sequence of the $h$-th human. We apply 4D-Humans (Goel et al., 2023) to each camera's video independently in our multi-camera setup. This yields separately processed results of human tracklets and poses: $\hat{\mathbf{T}} = \bigcup_{c=0}^{C-1} \hat{\mathbf{T}}^c$ and $\hat{\boldsymbol{\theta}} = \bigcup_{c=0}^{C-1} \hat{\boldsymbol{\theta}}^c$, where $\hat{\mathbf{T}}^c = \{\hat{\mathbf{T}}_j^c\}_{j \in \mathcal{D}^c}$ and $\hat{\boldsymbol{\theta}}^c = \{\hat{\boldsymbol{\theta}}_j^c\}_{j \in \mathcal{D}^c}$ represent the predicted tracklets and poses from camera $c$, respectively. Here, $\mathcal{D}^c$ is the set of detected human indices in camera $c$. Our task is to reconstruct $\boldsymbol{\theta}$ using $\hat{\boldsymbol{\theta}}$. We achieve this through the following steps:

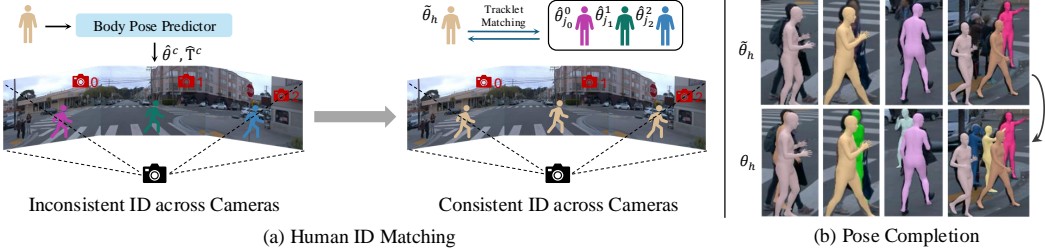

|  |  |
| :---: | :---: |
| (a) Human ID Matching | (b) Pose Completion |

Figure 3: **Human Pose Processing.** (a) Human ID matching ensures consistent identification across cameras. (b) Missing pose completion to recover poses of occluded individuals.

**Tracklet Matching:** We define a matching function $\mathcal{M}$ that finds the most similar predicted tracklets for each ground truth tracklet by computing the maximum mean IoU of their 2D projections:

$$\tilde{\boldsymbol{\theta}}_h = \mathcal{M}(h, \hat{\boldsymbol{\theta}}, \mathbf{T}, \hat{\mathbf{T}}). \tag{5}$$

This function learns a matching between ground truth tracklets and predicted tracklets, then outputs the corresponding matched pose sequences. Consider 3-camera setup as an example (Fig. 3(a)), if the $h$-th ground truth tracklet matches with detected tracklets $j_0, j_1, j_2$ in cameras 0 to 2 respectively, then $\tilde{\boldsymbol{\theta}}_h = \{\hat{\boldsymbol{\theta}}_{j_0}^0, \hat{\boldsymbol{\theta}}_{j_1}^1, \hat{\boldsymbol{\theta}}_{j_2}^2\}$, where $\hat{\boldsymbol{\theta}}_{j_k}^c$ is the pose sequence from camera $c$ for the detected tracklet $j_k$.

**Pose Completion:** As visualized in Fig. 3(b), 4D-Humans (Goel et al., 2023) fails to predict SMPL poses for occluded individuals in driving scenes, we design a process to recover missing poses:

$$\boldsymbol{\theta}_h = \mathcal{H}(\tilde{\boldsymbol{\theta}}_h, \mathbf{T}, \hat{\mathbf{T}}). \tag{6}$$

Here, function $\mathcal{H}$ identifies missing detections by comparing the ground truth and predicted tracklets, and interpolates missing poses to complete $\boldsymbol{\theta}_h$ from $\tilde{\boldsymbol{\theta}}_h$.

### 4.3 OPTIMIZATION

We simultaneously optimize all the parameters as mentioned in § 4.1 in *a single stage* to reconstruct the entire scene. These parameters include: **(1)** all the Gaussian attributes (opacity, mean positions, scaling, rotation, and appearance) in their local spaces, namely $\mathcal{G}^{\text{bg}}, \{\bar{\mathcal{G}}_v^{\text{rigid}}\}, \{\bar{\mathcal{G}}_h^{\text{SMPL}}\}, \{\bar{\mathcal{G}}_h^{\text{deform}}\}$, **(2)** the poses of both rigid and non-rigid nodes for each frame $t$, i.e., $\{\mathbf{T}_v(t)\}, \{\mathbf{T}_h(t)\}$, **(3)** the human poses of all the SMPL nodes for each frame $t$, i.e., $\{\boldsymbol{\theta}(t)\}$, and the corresponding skinning weights, **(4)** the weight $\varphi$ of the deformation network $\mathcal{F}$, **(5)** the weight of the sky model.

We use the following objective function for optimization:

$$\mathcal{L} = (1 - \lambda_r)\,\mathcal{L}_1 + \lambda_r \mathcal{L}_{\text{SSIM}} + \lambda_{\text{depth}}\mathcal{L}_{\text{depth}} + \lambda_{\text{opacity}}\mathcal{L}_{\text{opacity}} + \mathcal{L}_{\text{reg}}, \tag{7}$$

where $\mathcal{L}_1$ and $\mathcal{L}_{\text{SSIM}}$ are the L1 and SSIM losses on rendered images, $\mathcal{L}_{\text{depth}}$ compares the rendered depth of Gaussians with sparse depth signals from LiDAR, $\mathcal{L}_{\text{opacity}}$ encourages the opacity of the Gaussians to align with the non-sky mask, and $\mathcal{L}_{\text{reg}}$ represents various regularization terms applied to different Gaussian representations. Detailed descriptions of loss terms are provided in the Appendix.

## 5 EXPERIMENTS

**Dataset.** We conduct experiments on the Waymo Open Dataset (Sun et al., 2020), which comprises real-world driving logs. We tested up to 32 dynamic scenes in Waymo, including eight highly complex dynamic scenes that, in addition to typical vehicles, also contain diverse dynamic classes such as pedestrians and cyclists. Each selected segment contains approximately 150 frames. The segment IDs are listed in Tab. 12 and Tab. 6. To further demonstrate our effectiveness on common driving scenes, we extend our results to 5 additional popular driving datasets: NuScenes (Caesar et al., 2020), Argoverse2 (Wilson et al., 2023), PandaSet (Xiao et al., 2021), KITTI (Geiger et al., 2012), and NuPlan (Caesar et al., 2021).

**Baselines.** We compare our method against several Gaussian Splatting approaches: 3DGS (Kerbl et al., 2023), DeformableGS (Yang et al., 2023c), StreetGS (Yan et al., 2024), HUGS (Zhou et al.,

Table 1: **Comparison on Waymo Open Dataset.** We compute PSNR and SSIM for both the full image and dynamic regions. *Vehicle* indicates regions corresponding to vehicle-related classes, while *Human* indicates regions corresponding to human-related classes. *Box* indicates methods that utilize bounding boxes for dynamic modeling. *LiDAR* means method using LiDAR information.

| | | | Scene Reconstruction | | | | | | Novel View Synthesis | | | | | |
| | | | Full Image | | Human | | Vehicle | | Full Image | | Human | | Vehicle | |
| Methods | Box | LiDAR | PSNR↑ | SSIM↑ | PSNR↑ | SSIM↑ | PSNR↑ | SSIM↑ | PSNR↑ | SSIM↑ | PSNR↑ | SSIM↑ | PSNR↑ | SSIM↑ |
|---|---|---|---|---|---|---|---|---|---|---|---|---|---|---|
| EmerNeRF(Yang et al., 2023a) | | ✓ | 31.93 | 0.902 | 22.88 | 0.578 | 24.65 | 0.723 | 29.67 | 0.883 | 20.32 | 0.454 | 22.07 | 0.609 |
| 3DGS(Kerbl et al., 2023) | | ✓ | 26.00 | 0.912 | 16.88 | 0.414 | 16.18 | 0.425 | 25.57 | 0.906 | 16.62 | 0.387 | 16.00 | 0.407 |
| DeformGS(Yang et al., 2023c) | | ✓ | 28.40 | 0.929 | 17.80 | 0.460 | 19.53 | 0.570 | 27.72 | 0.922 | 17.30 | 0.426 | 18.91 | 0.530 |
| PVG(Chen et al., 2023) | | ✓ | 32.37 | 0.937 | 24.06 | 0.703 | 25.02 | 0.787 | 30.19 | 0.919 | 21.30 | 0.567 | 22.28 | 0.679 |
| HUGS(Zhou et al., 2024) | ✓ | ✓ | 28.26 | 0.923 | 16.23 | 0.404 | 24.31 | 0.794 | 27.65 | 0.914 | 15.99 | 0.378 | 23.27 | 0.748 |
| StreetGS(Yan et al., 2024) | ✓ | ✓ | 29.08 | 0.936 | 16.83 | 0.420 | 27.73 | 0.880 | 28.54 | 0.928 | 16.55 | 0.393 | 26.71 | 0.846 |
| Ours | ✓ | ✓ | **34.25** | **0.954** | **28.15** | **0.845** | **28.91** | **0.892** | **32.57** | **0.942** | **24.36** | **0.727** | **27.57** | **0.858** |

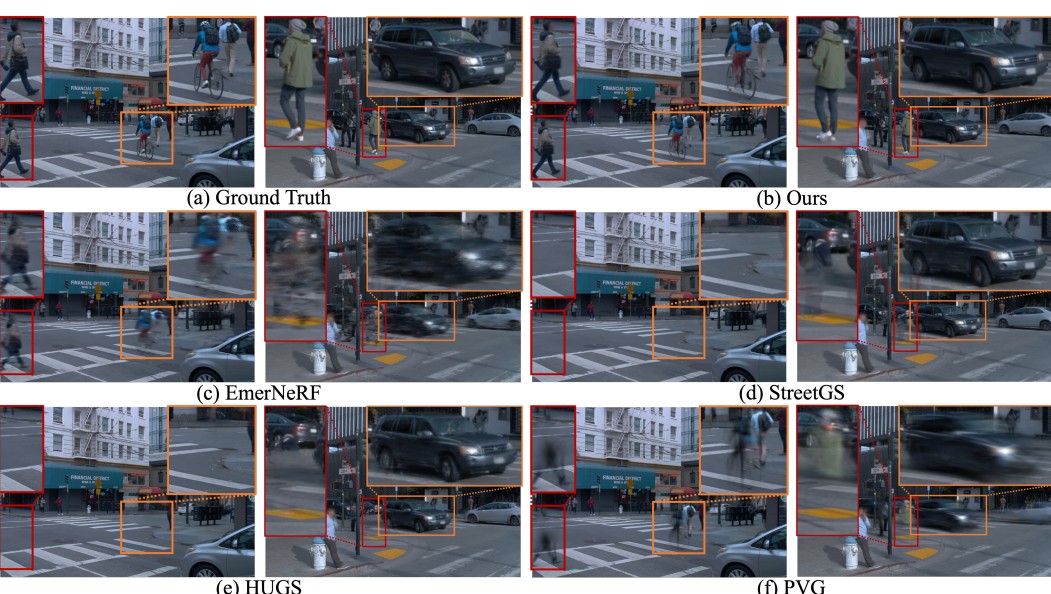

(a) Ground Truth  (b) Ours

(c) EmerNeRF  (d) StreetGS

(e) HUGS  (f) PVG

Figure 4: **Qualitative Comparison of Novel View Synthesis.** The insets highlight the details of the reconstructed dynamic objects. `OmniRe` manages to recover very fine details, achieving high-quality reconstruction of various common dynamic objects, including vehicles, pedestrians, and cyclists.

2024), and PVG (Chen et al., 2023). Additionally, we compare our method with NeRF-based approach EmerNeRF (Yang et al., 2023a). Among methods compared, for StreetGS (Yan et al., 2024), we use our own reimplementation. For 3DGS (Kerbl et al., 2023) and DeformableGS (Yang et al., 2023c), we use the implementation with LiDAR supervision to ensure the comparison fairness. For other methods, we use their official code. For training, we utilize data from the three front-facing cameras, resized to a resolution of 640×960 for all methods, along with LiDAR data for supervision. We utilize the instance bounding boxes provided by the dataset to transform objects and refine them via pose optimization during training. For further implementation details, please refer to Appendix.

## 5.1 MAIN RESULTS

**Appearance.** We evaluate our method on scene reconstruction and novel view synthesis (NVS) tasks, using every 10th frame as the held-out test set for NVS. We report PSNR and SSIM scores for full images, as well as human-related and vehicle-related regions, to assess dynamic reconstruction capabilities. The quantitative results in Tab. 1 show that `OmniRe` outperforms all other methods, with a significant margin in human-related regions, validating our holistic modeling of dynamic actors. Additionally, while StreetGS (Yan et al., 2024) and our method model vehicles in a similar way, we observe that `OmniRe` is slightly better than StreetGS even in vehicle regions. This is due to the absence of human modeling in StreetGS, which allows supervision signals from human regions (e.g., colors, LiDAR depth) to incorrectly influence vehicle modeling. The issues StreetGS faces are one of our motivations for modeling almost everything in a scene holistically, aiming to eliminate erroneous supervision and unintended gradient propagation.

In addition, we show visualizations in Fig. 4 to assess model performance qualitatively. Although PVG (Chen et al., 2023) performs well on the scene reconstruction task, it struggles with the

novel view synthesis task in highly dynamic scenes, resulting in blurry dynamic objects in novel views (Fig. 4-(f)). HUGS (Zhou et al., 2024) (Fig. 4-(e)), StreetGS (Yan et al., 2024)(Fig. 4-(d)) and 3DGS (Kerbl et al., 2023) (Fig. 8-(h)) fail to recover the pedestrians because they are not capable of modeling non-rigid objects. DeformableGS (Yang et al., 2023c) (Fig. 8-(g)) suffers from extreme motion blur for outdoor dynamic scenes with rapid movements, despite achieving reasonable performance for indoor scenes and cases with small motion. EmerNeRF (Yang et al., 2023a) reconstructs coarse structures of moving humans and vehicles to a certain level, but struggles with fine-grained details(Fig. 4-(c)). In contrast to all these methods in comparison, our method faithfully reconstructs fine details for any object in the scene, handling occlusion, deformation, and extreme motion. Video comparisons are included in the project page.

**Geometry.** In addition to appearance, we also investigate whether our method can reconstruct fine geometry of urban scenes. We evaluate the Root Mean Squared Error (RMSE) and two-way Chamfer Distances (CD) for LiDAR depth reconstruction on both training frames and novel frames. Details about evaluation procedures are provided in the Appendix. Tab. 3 reports the results. Our method outperforms others by a large margin. Fig. 5 illustrates the accurate reconstruction of dynamic actors achieved by our method in comparison to other approaches.

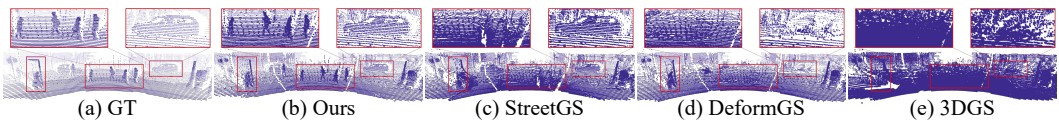

| (a) GT | (b) Ours | (c) StreetGS | (d) DeformGS | (e) 3DGS |

Figure 5: **Visualizations of Rendered LiDAR.** Our method accurately reconstructs LiDAR data for humans and vehicles compared to other approaches.

## 5.2 ABLATION STUDIES & APPLICATIONS

**SMPL Modeling.** SMPL modeling is important to model the local, continuous movements of humans. We study its impact by disabling the human pose transformation enabled by SMPL and report the results in Tab. 2 ((a) v.s. (b)) and illustrate these effects in Fig. 7-(B). Without template-based modeling, the reconstructed human renderings appear highly blurred, particularly around the legs, thus failing to accurately reconstruct human body movements. This contrasts sharply with the precise leg reconstruction observed in our default setting. Moreover, SMPL modeling provides joint-level control, improving the controllability (Fig. 1-(c,3), (c,4)).

**Human Body Pose Refinement.** The human body poses extracted as described in (§ 4.2) exhibit prediction errors and scale ambiguity, which subsequently lead to pose errors that degrade reconstruction quality, as shown in Fig. 6 (Noisy). We improve this by jointly optimizing the human poses and Gaussians via the same reconstruction losses. Tab. 2-(a) v.s. (c) ablates this design choice, and Fig. 6 showcases the refined poses. These results verify the effectiveness of our refinement strategy.

**Deformable Nodes.** Deformable nodes are important for accurately reconstructing out-of-distribution or template-less actors. Our approach addresses this challenge by learning a self-supervised deformation field that transforms Gaussians from their canonical space to the shape space. Tab. 2 ((a) v.s. (d)) proves the importance of this component. In Fig. 7-(A) shows that without deformable nodes, some dynamic actors are either ignored or incorrectly blended into the background.

**Boxes Refinement.** In practice, we observe that the instance bounding boxes provided by the dataset are imprecise. These noisy ground truth boxes can be harmful to rendering quality. To address this, we jointly refine the bounding box parameters during training. Tab. 4 and Fig. 12 show the practical

Table 2: **Ablation on Non-Rigid Modeling.**

|  | Full PSNR | | Human PSNR | |
| --- | --- | --- | --- | --- |
|  | Recon. | NVS | Recon. | NVS |
| (a) Ours default | **34.25** | **32.57** | **28.15** | **24.36** |
| (b) w/o SMPL actors | 32.80 | 31.76 | 24.71 | 23.18 |
| (c) w/o Body pose refine | 33.84 | 32.44 | 26.97 | 24.04 |
| (d) w/o Deformed actors | 33.64 | 32.17 | 25.26 | 22.41 |

Table 3: **Evaluation of LiDAR Depth Accuracy.**

|  | Training Frames | | Novel Frames | |
| --- | --- | --- | --- | --- |
| Methods | CD↓ | RMSE↓ | CD↓ | RMSE↓ |
| 3DGS(Kerbl et al., 2023) | 0.415 | 2.804 | 0.467 | 2.896 |
| DeformableGS(Yang et al., 2023c) | 0.384 | 2.965 | 0.383 | 2.990 |
| StreetGS(Yan et al., 2024) | 0.274 | 2.199 | 0.286 | 2.228 |
| Ours | **0.242** | **1.894** | **0.244** | **1.909** |

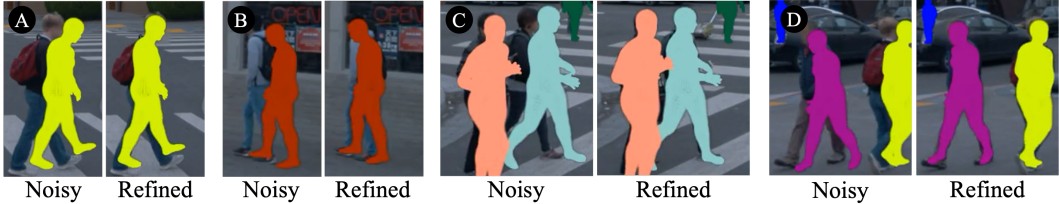

Figure 6: **Ablation of Human Body Pose Refinement.**

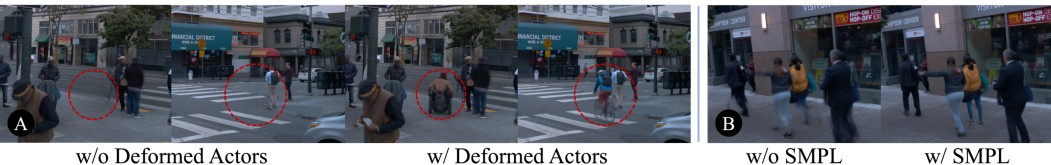

Figure 7: **Ablation of Human Modeling.**

benefits of this simple yet effective step, which results in improved numeric metrics and reduced blurriness of foreground objects.

**Applications to Simulation.** Thanks to the decomposition nature of `OmniRe`, each instance is modeled separately. After joint training, we obtain reconstructed assets that can be flexibly edited in terms of position and rotation. Beyond editing within a single scene, we can also transfer assets from one scene to another, adding variety and complexity to the reconstructed environments. Fig. 1(c,left) demonstrates a swap of the black vehicle originally in the scene (inset) with a reconstructed vehicle from another scene; and (c,right) an insertion of a pedestrian from the scene in the inset to a new scene. Additional car swap edits are shown in Fig. 11. Through explicit modeling of pedestrians and other non-rigid individuals, we achieve the simulation of reenacted scenarios involving detailed pedestrian-vehicle interaction. As demonstrated in Fig. 9, we simulate a moving vehicle stopping at a crossing, waiting for a pedestrian who slowly crosses. The pedestrian is reconstructed from another scene. This simulation of humans is extremely challenging for previous methods. This level of precise control over reconstructed photorealistic assets opens up possibilities for integration with previous simulators for automated simulation (Wang et al., 2022; Wei et al., 2024)

Table 4: **Ablation on GT Boxes Refinement.**

|  | Full PSNR | | Human PSNR | | Vehicle PSNR | |
|---|---|---|---|---|---|---|
|  | Recon. | NVS | Recon. | NVS | Recon. | NVS |
| Complete Model | **34.25** | **32.57** | **28.15** | **24.36** | **28.91** | **27.57** |
| w/o Box Refine. | 33.04 | 31.72 | 26.53 | 23.67 | 25.57 | 24.78 |

## 6 CONCLUSION

Our method, `OmniRe`, tackles comprehensive urban scene modeling using Gaussian Scene Graphs. It achieves fast, high-quality reconstruction and rendering, suggesting promise for driving and robotics simulation. We also present solutions for human modeling in complex environments. Future work includes self-supervised learning, improved scene representations, and safety/privacy considerations. To ensure reproducibility, the code is available at link.

**Broader impact.** Our method aims to address a significant problem in autonomous driving—simulation. This approach has the potential to enhance the development and testing of autonomous vehicles, potentially leading to safer and more efficient AV systems. Simulation, in a safe and controllable manner, remains an open and challenging research question.

**Limitations.** While enabling holistic scene modeling, `OmniRe` still has certain limitations. First, our method does not explicitly model lighting effects, which may lead to visual harmony issues during simulations, particularly when combining elements reconstructed under varying lighting conditions. Addressing this non-trivial challenge requires dedicated efforts beyond the scope of our current work. Further research into modeling light effects and enhancing simulation realism remains crucial for achieving more convincing and harmonious results. Second, similar to other per-scene optimization methods, `OmniRe` produces less satisfactory novel views when the camera deviates significantly from the training trajectories. Future works to address this issue include incorporating data-driven priors, such as image or video generative models, and optimizing camera poses jointly.

## 7 ETHICS STATEMENT

Our work does not involve the collection or annotation of new data. We utilize well-established public datasets that adhere to strict ethical guidelines. These datasets ensure that sensitive information, including identifiable human features, is blurred or anonymized to protect individual privacy. We are committed to ensuring that our method, as well as future applications, are employed responsibly and ethically to maintain safety and preserve privacy.

## 8 ACKNOWLEDGEMENTS

This work is supported by funding from Toyota Research Institute, Dolby, and Google DeepMind. Yue Wang is also supported by a Powell Faculty Research Award. We also thank Jiageng Mao, Junjie Ye, Ziyi Yang, Haozhe Lou, and Yifan Lu for their valuable discussions during the project, which helped us resolve issues and improve the methods.

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

# Supplemental Material

## A IMPLEMENTATION DETAILS

**Initialization:** For the background model, we refer to PVG (Chen et al., 2023), combining $6 \times 10^5$ LiDAR points with $4 \times 10^5$ random samples, which are divided into $2 \times 10^5$ near samples uniformly distributed by distance to the scene's origin and $2 \times 10^5$ far samples uniformly distributed by inverse distance. To initialize the background, we filter out the LiDAR samples of dynamic objects. For rigid nodes and non-rigid deformable nodes, we utilize their bounding boxes to accumulate the LiDAR points, while for non-rigid SMPL nodes, we initialize the Gaussians on the template mesh in their canonical space. To determine the initial color of Gaussians, we project LiDAR points onto the image plane, whereas random samples are initialized with random colors. The initial human body pose sequences of non-rigid SMPL Nodes are obtained through the process described in § 4.2.

**Training:** Our method trains for 30,000 iterations with all scene nodes optimized jointly. The learning rate for Gaussian properties aligns with the default settings of 3DGS (Kerbl et al., 2023), but varies slightly across different node types. Specifically, we set the learning rate for the rotation of Gaussians to $5 \times 10^{-5}$ for non-rigid SMPL nodes and $1 \times 10^{-5}$ for other nodes. The degrees of spherical harmonics are set to 3 for background nodes, rigid nodes, and non-rigid deformable nodes, while it is set to 1 for non-rigid SMPL nodes. The learning rate for the rotation of instance boxes is $1 \times 10^{-5}$, decreasing exponentially to $5 \times 10^{-6}$. The learning rate for the translation of instance boxes is $5 \times 10^{-4}$, decreasing exponentially to $1 \times 10^{-4}$. The learning rate for human body poses of non-rigid SMPL nodes is $5 \times 10^{-5}$, decreasing exponentially to $1 \times 10^{-5}$. For the Gaussian densification strategy, we utilize the absolute gradient of Gaussians introduced in Ye et al. (2024) to control memory usage. We set the densification threshold of position gradient to $3 \times 10^{-4}$. This use of absolute gradient has a minimal impact on performance, as discussed in detail in Appendix D.4. The densification threshold for scaling is $3 \times 10^{-3}$. Our method runs on a single NVIDIA RTX 4090 GPU, with training for each scene taking about 1 hour. Training time varies with different training settings.

**Optimization:** We utilize the loss function introduced in Eq (7) to jointly optimize all learnable parameters. The image loss is computed as:

$$\mathcal{L}_{\text{image}} = (1 - \lambda_r) \mathcal{L}_1 + \lambda_r \mathcal{L}_{\text{SSIM}} \tag{8}$$

due to sparse temporal-spatial observation of the dynamic part, its supervision signal is insufficient. To address this, we apply a higher image loss weight to the dynamic regions identified by the rendered dynamic mask. This weight is set to 5. The depth map loss is computed as:

$$\mathcal{L}_{\text{depth}} = \frac{1}{hw} \sum \left\| \mathcal{D}^s - \hat{\mathcal{D}} \right\|_1 \tag{9}$$

where $\mathcal{D}^s$ is the inverse of the sparse depth map. We project LiDAR points onto the image plane to generate the sparse LiDAR map, and $\hat{\mathcal{D}}$ is the inverse of the predicted depth map.

The mask loss $\mathcal{L}_{\text{opacity}}$ is computed as:

$$\mathcal{L}_{\text{opacity}} = -\frac{1}{hw} \sum O_{\mathcal{G}} \cdot \log O_{\mathcal{G}} - \frac{1}{hw} \sum M_{\text{sky}} \cdot \log(1 - O_{\mathcal{G}}) \tag{10}$$

where $M_{\text{sky}}$ is the sky mask, and $O_{\mathcal{G}}$ is the rendered opacity map.

In addition to the reconstruction losses, we introduce various regularization terms for different Gaussian representations to improve quality. Among these, an important regularization term is $\mathcal{L}_{\text{pose}}$, designed to ensure smooth human body poses $\boldsymbol{\theta}(t)$. This term is defined as:

$$\mathcal{L}_{\text{pose}} = \frac{1}{2} \left\| \boldsymbol{\theta}(t - \delta) + \boldsymbol{\theta}(t + \delta) - 2\boldsymbol{\theta}(t) \right\|_1 \tag{11}$$

where $\delta$ is a randomly chosen integer from $\{1, 2, 3, 4, 5\}$. We set the weight of the SSIM loss, $\lambda_r$, to 0.2, the depth loss, $\lambda_{\text{depth}}$, to 0.1, the opacity loss, $\lambda_{\text{opacity}}$, to 0.05, and the pose smoothness loss, $\lambda_{\text{pose}}$, to 0.01.

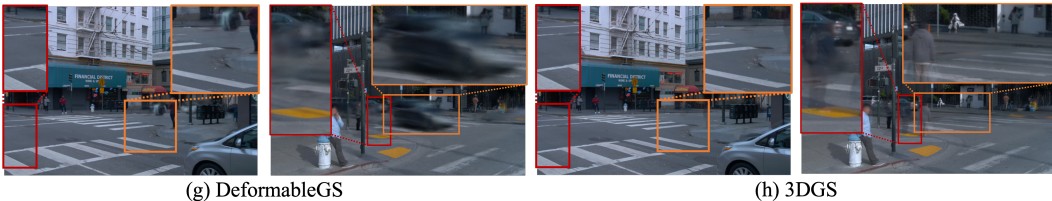

(g) DeformableGS        (h) 3DGS

Figure 8: Additional Qualitative Comparison of Novel View Synthesis.

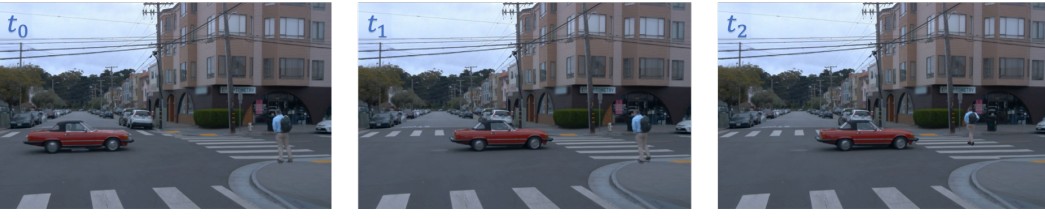

Figure 9: A sample of human-vehicle interaction simulation in driving scenarios.

## B  BASELINES

• **EmerNeRF** (Yang et al., 2023a) is a state-of-the-art NeRF-Based method for dynamic driving scene reconstruction. EmerNeRF uses a static field represented by a 3D Hash-Grid to model the static parts of the scene and a dynamic field with a 4D Hash-Grid to model the dynamic parts. Additionally, it employs a flow field to aggregate the dynamic features. This self-supervised decomposition approach yields good results on dynamic scene modeling and static-dynamic decomposition, with the scene flow emerging in the flow field.

• **DeformableGS** (Yang et al., 2023c) defines a canonical space to represent scenes with Gaussians. To model dynamics, it uses a deformation network to predict offsets of Gaussian properties. These offsets then deform the Gaussians to fit the scene dynamics. DeformableGS works well in synthetic and indoor datasets. We compare it to our method to evaluate its ability to model challenging out-door dynamic scenes.

• **StreetGS** (Yan et al., 2024) is a dynamic scene modeling method based on Gaussian Splatting for driving scenes. StreetGS models the components of dynamic scenes separately: the static background and the foreground vehicles. It utilizes boxes predicted by an off-the-shelf model to warp the Gaussians of foreground vehicles and refine them during training. StreetGS yields good results on driving scenes but ignores other non-rigid dynamic objects in the scene.

• **HUGS** (Zhou et al., 2024) is a GS-based method for driving scene modeling and understanding. It not only models the appearance of a scene but also distills 2D flow maps and semantic maps into the 3D scene to enable holistic urban scene understanding. Similar to StreetGaussian (Yan et al., 2024), HUGS uses object boxes for compositing dynamic elements. HUGS achieves good performance in both scene modeling and semantic modeling. However, it primarily focuses on rigid backgrounds and objects, without addressing non-rigid dynamics.

• **PVG** (Chen et al., 2023) introduces Periodic Vibration Gaussians that vibrate over time with optimizable vibration directions, life span, and life peak (the moment of highest opacity) to represent dynamic scenes. These Gaussians are optimized using a self-supervised approach. The method achieves static-dynamic decomposition by categorizing Gaussians based on their life spans. We compare PVG with our method to evaluate our capability in modeling highly complex dynamic scenes.

Among all the compared methods, HUGS (Zhou et al., 2024) and StreetGaussians (Yan et al., 2024) require bounding boxes of foreground objects. PVG (Chen et al., 2023), StreetGaussians (Yan et al., 2024), and EmerNeRF (Yang et al., 2023a) utilize LiDAR data for depth supervision. While the original implementations of 3DGS (Kerbl et al., 2023) and DeformableGS (Yang et al., 2023c) do not include depth supervision, we added LiDAR depth supervision in experiments to ensure a fair comparison.

## C  EVALUATION

**Appearance.** For the Novel View Synthesis task, we select every 10th frame from the original sequence as the test set. We use PSNR and SSIM to evaluate the quality of the rendered images. Since we focus on dynamic scenes, we also compute the PSNR and SSIM for regions with vehicles and humans. To identify regions of vehicles and humans, we use Segformer (Xie et al., 2021) to obtain semantic masks. We further identify the movable dynamic parts using projections of moving object bounding boxes, utilizing their velocity information. One example of dynamic masks cam bee seen in Fig. 10.

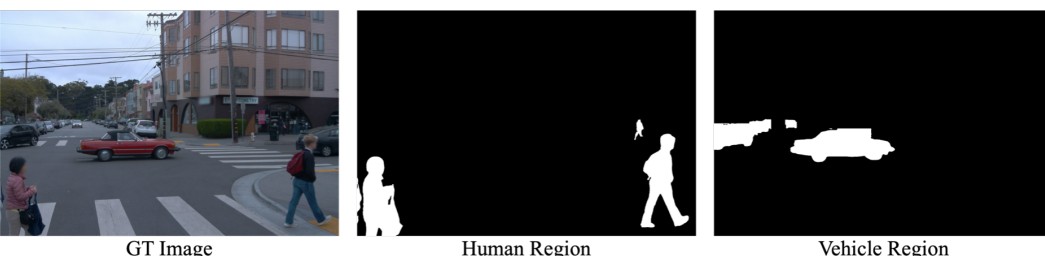

|  GT Image  |  Human Region  |  Vehicle Region  |

Figure 10: An example of the dynamic masks for computing dynamic region metrics.

**Geometry.** Our method uses LiDAR data to initialize Gaussians and supervise scene depth by comparing the rendered depth map with the sparse LiDAR depth map. Post-training, Gaussians typically deviate from their initial state through densification or optimization, Therefore, comparing the LiDAR depth reconstruction is still a valid comparison. We follow the depth evaluation method of StreetSurf (Guo et al., 2023): render a depth map and match depth pixels to LiDAR rays. For Chamfer Distance, re-project the predicted depth to 3D using the LiDAR ray direction and origin. For RMSE, compare the GT and predicted ranges for LiDAR rays.

## D  ADDITIONAL RESULTS

### D.1  QUALITATIVE COMPARISON

We recommend readers to our project page for video comparisons of the methods.

### D.2  QUANTITATIVE COMPARISON

To further validate our method's effectiveness, we tested our method against StreetGS (Yan et al., 2024) and EmerNeRF (Yang et al., 2023a) on 32 dynamic scenes from the Waymo dataset, with results reported in Tab. 5.

Table 5: We expanded our evaluation to 32 dynamic scenes from the Waymo dataset, comparing our method with StreetGS (Yan et al., 2024) and EmerNeRF (Yang et al., 2023a). The segment IDs are listed in Tab. 6.

| | Scene Reconstruction | | | | | | Novel View Synthesis | | | | | |
| | Full Image | | Human | | Vehicle | | Full Image | | Human | | Vehicle | |
| Methods | PSNR↑ | SSIM↑ | PSNR↑ | SSIM↑ | PSNR↑ | SSIM↑ | PSNR↑ | SSIM↑ | PSNR↑ | SSIM↑ | PSNR↑ | SSIM↑ |
|---|---|---|---|---|---|---|---|---|---|---|---|---|
| EmerNeRF | 31.29 | 0.877 | 23.14 | 0.581 | 24.47 | 0.709 | 29.04 | 0.851 | 20.76 | 0.467 | 21.80 | 0.582 |
| StreetGS | 29.93 | 0.931 | 19.63 | 0.524 | 27.48 | 0.871 | 28.73 | 0.910 | 18.77 | 0.470 | 26.18 | 0.825 |
| Ours | **33.73** | **0.946** | **28.28** | **0.855** | **28.02** | **0.880** | **31.71** | **0.924** | **24.57** | **0.730** | **26.55** | **0.833** |

Table 6: Segment IDs of 32 dynamic scenes of Waymo Dataset used in the test for Tab. 5.

| | | | | | | | |
|---|---|---|---|---|---|---|---|
| seg104554... | seg125050... | seg169514... | seg584622... | seg776165... | seg138251... | seg448767... | seg965324... |
| seg119252... | seg122514... | seg132544... | seg134024... | seg166004... | seg173881... | seg215148... | seg391164... |
| seg454855... | seg560223... | seg571325... | seg587066... | seg842457... | seg952165... | seg952995... | seg112365... |
| seg152664... | seg411445... | seg123218... | seg102252... | seg148106... | seg265611... | seg179934... | seg104859... |

## D.3 OMNIRE ON CHALLENGING SCENES

To demonstrate the effectiveness and robustness of OmniRe, we tested our method alongside StreetGS (Yan et al., 2024), PVG (Chen et al., 2023), and DeformableGS (Yang et al., 2023c) on various challenging scenes. The results show that OmniRe maintains high reconstruction quality under most challenging conditions. This section provides the quantitative performance of the compared methods. Video visualizations and comparison are included in our project page.

**Super Crowded Scenes.** Reconstructing scenes with extreme dynamic occlusions poses significant challenges. OmniRe is specifically designed to handle these scenes and performs well even in highly occluded scenes. We conducted experiments on three extremely crowded scenes from Waymo (seg112520..., seg104859..., seg152664...), including situations with a large crowd of people simultaneously crossing a street. Results in Tab. 7 show that OmniRe demonstrates strong performance in these scenes.

Table 7: Comparison on super crowded scenes.

| | Scene Reconstruction | | | | | | Novel View Synthesis | | | | | |
| | Full Image | | Human | | Vehicle | | Full Image | | Human | | Vehicle | |
| Methods | PSNR↑ | SSIM↑ | PSNR↑ | SSIM↑ | PSNR↑ | SSIM↑ | PSNR↑ | SSIM↑ | PSNR↑ | SSIM↑ | PSNR↑ | SSIM↑ |
|---|---|---|---|---|---|---|---|---|---|---|---|---|
| DeformGS | 26.69 | 0.861 | 19.48 | 0.504 | 17.49 | 0.455 | 25.66 | 0.839 | 18.71 | 0.448 | 17.12 | 0.414 |
| PVG | 29.11 | 0.865 | 24.56 | 0.686 | 21.29 | 0.644 | 27.21 | 0.830 | 22.24 | 0.555 | 19.63 | 0.528 |
| StreetGS | 26.81 | 0.865 | 19.31 | 0.516 | 24.79 | 0.787 | 25.67 | 0.841 | 18.53 | 0.459 | 23.40 | 0.723 |
| Ours | **31.25** | **0.900** | **27.50** | **0.809** | **25.91** | **0.812** | **28.91** | **0.866** | **23.78** | **0.675** | **24.46** | **0.750** |

**Nighttime Scenes.** We tested three nighttime scenes (seg129008..., seg102261..., seg128560...) from Waymo. Results in Tab. 8 indicate that OmniRe outperforms the compared methods, achieving superior reconstruction quality under low-light conditions.

Table 8: Comparison on night time scenes.

| | Scene Reconstruction | | | | | | Novel View Synthesis | | | | | |
| | Full Image | | Human | | Vehicle | | Full Image | | Human | | Vehicle | |
| Methods | PSNR↑ | SSIM↑ | PSNR↑ | SSIM↑ | PSNR↑ | SSIM↑ | PSNR↑ | SSIM↑ | PSNR↑ | SSIM↑ | PSNR↑ | SSIM↑ |
|---|---|---|---|---|---|---|---|---|---|---|---|---|
| DeformGS | 30.06 | 0.767 | 27.40 | 0.665 | 20.62 | 0.612 | 28.88 | 0.740 | 22.30 | 0.530 | 19.20 | 0.535 |
| PVG | 30.55 | 0.768 | 30.74 | 0.777 | 22.77 | 0.713 | 29.32 | 0.740 | 25.22 | 0.634 | 20.90 | 0.611 |
| StreetGS | 30.60 | 0.775 | 27.54 | 0.667 | 25.68 | 0.772 | 29.38 | 0.741 | 21.88 | 0.523 | **24.61** | **0.729** |
| Ours | **31.14** | **0.778** | **31.06** | **0.790** | **25.87** | **0.774** | **29.79** | **0.744** | **25.71** | **0.689** | 24.29 | 0.724 |

**Adverse Weather Conditions.** To evaluate performance under adverse weather conditions, we tested seven scenes with varying weather types: a. **rainy** (seg113555..., seg109277..., seg141339...) b. **foggy** (seg161022..., seg172163...) c. **cloudy** (seg144275..., seg157956...). OmniRe handled these challenging conditions robustly, maintaining high reconstruction fidelity across all weather conditions (Tab. 9).

Table 9: Comparison on adverse weather scenes.

| | Scene Reconstruction | | | | | | Novel View Synthesis | | | | | |
| | Full Image | | Human | | Vehicle | | Full Image | | Human | | Vehicle | |
| Methods | PSNR↑ | SSIM↑ | PSNR↑ | SSIM↑ | PSNR↑ | SSIM↑ | PSNR↑ | SSIM↑ | PSNR↑ | SSIM↑ | PSNR↑ | SSIM↑ |
|---|---|---|---|---|---|---|---|---|---|---|---|---|
| DeformGS | 32.92 | 0.933 | 20.06 | 0.497 | 23.12 | 0.658 | 31.43 | 0.916 | 19.65 | 0.478 | 22.06 | 0.598 |
| PVG | 32.75 | 0.927 | 27.75 | 0.785 | 27.20 | 0.795 | 31.12 | 0.907 | 25.02 | 0.656 | 24.44 | 0.677 |
| StreetGS | 33.49 | 0.933 | 18.26 | 0.424 | 32.28 | 0.916 | 32.05 | 0.918 | 18.56 | 0.434 | 29.60 | 0.841 |
| Ours | **34.00** | **0.935** | **30.12** | **0.857** | **32.55** | **0.919** | **32.58** | **0.920** | **26.75** | **0.763** | **29.82** | **0.844** |

**High-Speed Scenes.** For high-speed scenes such as highways, we tested three highway scenes from the Waymo dataset (seg109239..., seg113924..., seg118396). Results are provided in Tab. 10. As highways typically lack non-rigid objects (e.g., humans), human metrics were not applicable. Since non-rigid modeling was not required in these scenes, OmniRe and StreetGS demonstrated comparable performance.

Table 10: Comparison on high speed scenes.

| | Scene Reconstruction | | | | | | Novel View Synthesis | | | | | |
| | Full Image | | Human | | Vehicle | | Full Image | | Human | | Vehicle | |
| Methods | PSNR↑ | SSIM↑ | PSNR↑ | SSIM↑ | PSNR↑ | SSIM↑ | PSNR↑ | SSIM↑ | PSNR↑ | SSIM↑ | PSNR↑ | SSIM↑ |
|---|---|---|---|---|---|---|---|---|---|---|---|---|
| DeformGS | 26.95 | 0.855 | - | - | 17.49 | 0.482 | 25.56 | 0.837 | - | - | 16.07 | 0.407 |
| PVG | 28.28 | 0.861 | - | - | 21.14 | 0.617 | 25.79 | 0.825 | - | - | 17.65 | 0.470 |
| StreetGS | **30.89** | **0.902** | - | - | 29.45 | **0.883** | **28.05** | 0.863 | - | - | **24.88** | **0.766** |
| Ours | 30.85 | **0.902** | - | - | **29.51** | 0.882 | 28.01 | **0.864** | - | - | 24.52 | 0.760 |

### D.4 ABLATION STUDIES

**Absolute Gradient.** In our implementation, we applied AbsGrad for 3DGS densification across all reproduced methods (StreetGS, DeformableGS, and 3DGS) as a standard practice. To quantify the impact of AbsGrad, we conducted a comparative study with and without its application. The results of this analysis are presented in Tab. 11. We see that disabling AbsGrad leads to a marginal performance decrease (about 0.1 PSNR) for all methods, proving that AbsGrad is not the key factor contributing to our performance advantage over others. Note that DeformableGS fails to run due to out-of-memory issues when AbsGrad was disabled. Based on these findings, we recommend the incorporation of AbsGrad as a standard practice in 3DGS densification and related methods.

Table 11: **Ablation on AbsGrad.** By default, we apply AbsGrad to all GS-based approaches reproduced by us. We now disable it to analyze its impact. We mark methods with AbsGrad enabled with grey background. We observe that 1) DeformableGS fails under w/o. AbsGrad setting because of out of memory issue; 2) enabling AbsGrad is a good practice ( +0.1 PNSR for all methods) but not an enabling factor for our performance lead.

| | Scene Reconstruction | | | | | | Novel View Synthesis | | | | | |
| | Full Image | | Human | | Vehicle | | Full Image | | Human | | Vehicle | |
| Methods | PSNR↑ | SSIM↑ | PSNR↑ | SSIM↑ | PSNR↑ | SSIM↑ | PSNR↑ | SSIM↑ | PSNR↑ | SSIM↑ | PSNR↑ | SSIM↑ |
|---|---|---|---|---|---|---|---|---|---|---|---|---|
| EmerNeRF | 31.93 | 0.902 | 22.88 | 0.578 | 24.65 | 0.723 | 29.67 | 0.883 | 20.32 | 0.454 | 22.07 | 0.609 |
| 3DGS* | 26.00 | 0.912 | 16.88 | 0.414 | 16.18 | 0.425 | 25.57 | 0.906 | 16.62 | 0.387 | 16.00 | 0.407 |
| 3DGS | 25.84 | 0.910 | 16.69 | 0.405 | 16.02 | 0.415 | 25.61 | 0.905 | 16.52 | 0.383 | 15.97 | 0.405 |
| DeformGS* | 28.40 | 0.929 | 17.80 | 0.460 | 19.53 | 0.570 | 27.72 | 0.922 | 17.30 | 0.426 | 18.91 | 0.530 |
| DeformGS | – | – | – | – | – | – | – | – | – | – | – | – |
| PVG | 32.37 | 0.937 | 24.06 | 0.703 | 25.02 | 0.787 | 30.19 | 0.919 | 21.30 | 0.567 | 22.28 | 0.679 |
| HUGS | 28.26 | 0.923 | 16.23 | 0.404 | 24.31 | 0.794 | 27.65 | 0.914 | 15.99 | 0.378 | 23.27 | 0.748 |
| StreetGS* | 29.08 | 0.936 | 16.83 | 0.420 | 27.73 | 0.880 | 28.54 | 0.928 | 16.55 | 0.393 | 26.71 | 0.846 |
| StreetGS | 28.89 | 0.932 | 16.70 | 0.409 | 28.07 | 0.878 | 28.46 | 0.926 | 16.41 | 0.387 | 26.86 | 0.845 |
| Ours* | 34.25 | 0.954 | 28.15 | 0.845 | 28.91 | 0.892 | 32.57 | 0.942 | 24.36 | 0.727 | 27.57 | 0.858 |
| Ours | 34.11 | 0.953 | 28.00 | 0.842 | 28.83 | 0.890 | 32.46 | 0.941 | 24.28 | 0.726 | 27.55 | 0.857 |

Table 12: Segment IDs of 8 dynamic scenes of Waymo Dataset used in the test for Tab. 1. and Tab. 11

| seg104554... | seg125050... | seg169514... | seg584622... | seg776165... | seg138251... | seg448767... | seg965324... |
|---|---|---|---|---|---|---|---|

**Additional Results** The Tab. 14 is the full table of Tab. 2 that includes evaluation on SSIM. The Tab. 13 is the full table of Tab. 4 that includes evaluation on SSIM.

Table 13: **Ablation on GT Boxes Refinement.**

| | Scene Reconstruction | | | | | | Novel View Synthesis | | | | | |
| | Full Image | | Human | | Vehicle | | Full Image | | Human | | Vehicle | |
| | PSNR↑ | SSIM↑ | PSNR↑ | SSIM↑ | PSNR↑ | SSIM↑ | PSNR↑ | SSIM↑ | PSNR↑ | SSIM↑ | PSNR↑ | SSIM↑ |
|---|---|---|---|---|---|---|---|---|---|---|---|---|
| Complete Model | 34.25 | 0.954 | 28.15 | 0.845 | 28.91 | 0.892 | 32.57 | 0.942 | 24.36 | 0.727 | 27.57 | 0.858 |
| w/o Box Refine. | 33.04 | 0.947 | 26.53 | 0.790 | 25.57 | 0.813 | 31.72 | 0.936 | 23.67 | 0.686 | 24.78 | 0.785 |

## E OMNIRE IN PRACTICE

**Bounding Boxes.** Similar to other scene-graph-based approaches (Ost et al., 2021; Yang et al., 2023b; Tonderski et al., 2024; Fischer et al., 2024b; Zhou et al., 2023; 2024; Yan et al., 2024), we utilize bounding boxes for driving scene reconstruction, as they are widely used for producing superior reconstruction results compared to methods that do not employ them. Additionally, bounding boxes offer significant controllability. It allows precise manipulation of both rigid objects like vehicles and non-rigid objects such as individual human body movements—an ability lacking in self-supervised methods like EmerNeRF (Yang et al., 2023a) and PVG (Chen et al., 2023) that do not use instance information. This level of controllability is crucial for tasks like scene simulation, which require the ability to manage the movement of all participating agents. Lastly, bounding box annotation is a standard and generally straightforward process in the autonomous driving field, with most popular datasets already providing these annotations via established auto-labeling tools, thereby minimizing manual effort and making the resource both efficient and accessible. For real-world driving logs, these auto-labeling tools can generate precise bounding boxes at little cost.

Table 14: **Ablation on Non-Rigid Modeling.**

| | Scene Reconstruction | | | | Novel View Synthesis | | | |
| | Full Image | | Human | | Full Image | | Human | |
| | PSNR↑ | SSIM↑ | PSNR↑ | SSIM↑ | PSNR↑ | SSIM↑ | PSNR↑ | SSIM↑ |
|---|---|---|---|---|---|---|---|---|
| (a) Ours default | **34.25** | **0.954** | **28.15** | **0.845** | **32.57** | **0.942** | **24.36** | **0.727** |
| (b) w/o SMPL actors | 32.80 | 0.949 | 24.71 | 0.770 | 31.76 | 0.939 | 23.18 | 0.694 |
| (c) w/o Body Pose Refine. | 33.84 | 0.952 | 26.97 | 0.815 | 32.44 | 0.941 | 24.04 | 0.712 |
| (d) w/o Deformed actors | 33.64 | 0.953 | 25.26 | 0.766 | 32.17 | 0.941 | 22.41 | 0.653 |

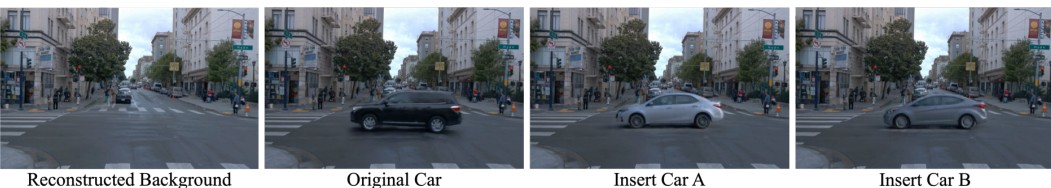

| Reconstructed Background | Original Car | Insert Car A | Insert Car B |

Figure 11: Our method allows for flexible editing of scene assets.

**How to Determine Gaussian Representations for Humans?** We categorize pedestrians into two groups for modeling. Near-range pedestrians, detected by our human pose processing module introduced in § 4.2, are modeled using SMPL nodes. Far-range pedestrians, typically undetected due to distance, are modeled using deformable nodes. This approach naturally distinguishes between near and far-range pedestrians based on human detection capabilities.

Other individuals, such as those using wheelchairs, skateboards, or bicycles, are often labeled as "cyclists" in the datasets we study. However, these labels may be specific to the dataset used, and in some cases, the annotations might not be accurate. For instance, in the Waymo Dataset, a person on a motorcycle may be labeled as a "vehicle". This reliance on dataset-specific labels could potentially limit the generalization of our method to other scenarios with imperfect labels.

To address this issue, we conducted preliminary experiments using GPT-4o (Achiam et al., 2023) to classify individuals (cropped by bounding boxes) into two categories: pedestrians and humans using personal transportation devices (e.g., wheelchairs, bicycles, motorcycles). Testing on 60 individuals (30 from each category), GPT-4o (Achiam et al., 2023) achieved 100% accuracy. This suggests that accurate labels can be obtained relatively easily, thanks to the development of vision-language models.

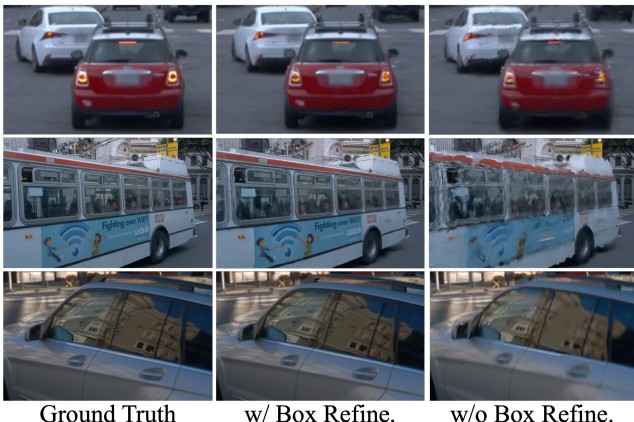

| Ground Truth | w/ Box Refine. | w/o Box Refine. |

Figure 12: **Ablation of Boxes Refinement.**

