# OpenReview forum: "OmniRe: Omni Urban Scene Reconstruction"
_ICLR.cc/2025/Conference — ICLR 2025 Spotlight_

### Official Review · Reviewer_h6SA · 2024-10-22

**Soundness:** 3
**Presentation:** 3
**Contribution:** 3
**Rating:** 8
**Confidence:** 4

**Summary:**

This paper introduces OmniRe, a comprehensive framework for dynamic urban scene reconstruction. It leverages neural scene graphs with Gaussian representations to unify the reconstruction of static backgrounds, moving vehicles, and non-rigidly dynamic actors. Additionally, it incorporates specialized designs for human modeling. The effectiveness of the approach is demonstrated across multiple datasets, showcasing superior performance in both reconstruction quality and novel view synthesis.

**Strengths:**

1. The paper is clearly written, with illustrative figures that are easy to understand. The experiments are comprehensive.
2. Modeling dynamic objects and simulating interactive behaviors are essential for closed-loop simulation in autonomous driving systems.
3. This work is highly engineering-oriented and demonstrates impressive results. Additionally, the authors have committed to open-sourcing the code, which will have significant value in advancing autonomous driving simulation in the future.

**Weaknesses:**

As mentioned by the authors in the limitations section, there are still two key shortcomings: 1. The lack of lighting modeling results in unnatural object insertions. 2. The synthesis of new viewpoints is constrained to the original trajectory, limiting the approach from achieving fully free-trajectory digital reconstruction.

**Questions:**

1. GS-based methods generally perform well in scenarios with static environments or low vehicle speeds, as demonstrated by most of the demos on the project page. However, I am curious about the reconstruction performance of this approach in situations where the ego vehicle is moving at higher speeds.
2. I wonder about the computational cost of reconstructing a complete segment in the Waymo dataset, as the entire pipeline seems a bit complex.
3. Why does it seem that the reconstruction quality of NuPlan is significantly worse than that of other datasets?

---

> ### Author Response · Authors · 2024-11-22
> **Response to Reviewer h6SA (1/2)**
>
> We sincerely thank the reviewer for the time and effort invested in reviewing our manuscript. We also appreciate the recognition of our work in **advancing driving reconstruction and simulation systems, as well as the engineering effort involved.** Your comments are very helpful, our responses are detailed below.
>
> > Q1. As mentioned by the authors in the limitations section, there are still two key shortcomings: 1. The lack of lighting modeling results in unnatural object insertions. 2. The synthesis of new viewpoints is constrained to the original trajectory, limiting the approach from achieving fully free-trajectory digital reconstruction.
>
> Thank you for your thoughtful comments on the limitations discussed in our paper. While addressing these challenges is non-trivial and slightly beyond the scope of our current work, we would like to propose a few potential directions to address them.
>
> For light modeling, potential directions include: 1) incorporate BRDF model to learn reflectance, material properties, and utilize environmental maps for illumination modeling; 2) post-processing rendered images with harmonization techniques, such as rendering feature maps and training a network to convert these maps into photorealistic RGB images. 3) using video diffusion models to refine the outputs, e.g. localized editing with diffusion models [a].
>
> For novel view rendering, the current quality meets the requirements for typical driving simulations (e.g., lane shifts or varying vehicle heights). However, limitations arise during free navigation with viewpoints far outside the training space. For future work, we believe methods using generative models like [b] could enhance reconstruction by inferring missing details and completing unseen regions.
>
> [a] Stable Video Diffusion: Scaling Latent Video Diffusion Models to Large Datasets
> [b] Streetscapes: Large-scale Consistent Street View Generation Using Autoregressive Video Diffusion. SIGGRAPH2024

---

> ### Author Response · Authors · 2024-11-22
> **Response to Reviewer h6SA (2/2)**
>
> > Q2. GS-based methods generally perform well in scenarios with static environments or low vehicle speeds, as demonstrated by most of the demos on the project page. However, I am curious about the reconstruction performance of this approach in situations where the ego vehicle is moving at higher speeds.
>
> Thank you for raising this concern. OmniRe does perform equally well for high-speed scenarios. We have included some demos where the ego vehicle is moving at higher speeds (e.g., Pandaset scene-012, Argoverse scene-037 in the anonymous page).
>
> To fully address this concern, we conducted additional tests of OmniRe, StreetGS, PVG, and DeformGS, on three scenes (seg119252…, seg122514…, seg152664…) **with moderately high vehicle speeds.** The results are detailed below, showcasing OmniRe's superior performance in full scene reconstruction and human modeling, while performing on par with StreetGS in vehicle modeling.
>
> | Method   | Recon Full PSNR/SSIM | Recon Human PSNR/SSIM | Recon Vehicle PSNR/SSIM | Novel Full PSNR/SSIM | Novel Human PSNR/SSIM | Novel Vehicle PSNR/SSIM |
> | :------- | :------------------- | :-------------------- | :---------------------- | :------------------- | :-------------------- | :---------------------- |
> | omnire   | **31.75**/**0.923**  | **28.95**/**0.861**   | 27.77/0.865             | **29.44**/**0.887**  | **24.70**/**0.719**   | **26.44**/**0.822**     |
> | streetgs | 31.40/0.920          | 24.79/0.737           | **27.90**/**0.869**     | 29.24/0.884          | 22.93/0.630           | 26.32/0.819             |
> | pvg      | 29.81/0.887          | 26.87/0.799           | 24.37/0.758             | 27.78/0.844          | 24.24/0.665           | 21.92/0.625             |
> | deformgs | 30.71/0.917          | 24.05/0.703           | 20.16/0.579             | 28.45/0.877          | 22.39/0.590           | 19.03/0.505             |
>
> **Furthermore, we tested them on three extremely high-speed driving scenes** (seg109239…, seg113924…, seg118396) from Waymo. Highways typically lack non-rigid classes (e.g., humans), making human reconstruction metrics inapplicable. Since non-rigid modeling is not required in these scenarios, OmniRe and StreetGS demonstrate comparable performance. **We provide additional videos in our [anonymous page](https://anonymousi079j.github.io/omnire_id4816/) (Section: OmniRe on Challenging Scenes).** We will greatly appreciate it if reviewers get a chance to check these additional results of highly challenging scenes.
>
> | Method   | Recon Full PSNR/SSIM | Recon Human PSNR/SSIM | Recon Vehicle PSNR/SSIM | Novel Full PSNR/SSIM | Novel Human PSNR/SSIM | Novel Vehicle PSNR/SSIM |
> | :------- | :------------------- | :-------------------- | :---------------------- | :------------------- | :-------------------- | :---------------------- |
> | omnire   | 30.85/**0.902**      | -/-                   | **29.51**/0.882         | 28.01/**0.864**      | -/-                   | 24.52/0.760             |
> | streetgs | **30.89**/**0.902**  | -/-                   | 29.45/**0.883**         | **28.05**/0.863      | -/-                   | **24.88**/**0.766**     |
> | pvg      | 28.28/0.861          | -/-                   | 21.14/0.617             | 25.79/0.825          | -/-                   | 17.65/0.470             |
> | deformgs | 26.95/0.855          | -/-                   | 17.49/0.482             | 25.56/0.837          | -/-                   | 16.07/0.407             |
>
> > Q3. I wonder about the computational cost of reconstructing a complete segment in the Waymo dataset, as the entire pipeline seems a bit complex.
>
> Our pipeline is cleanly organized as shown in the supplement material. We have dedicated significant engineering efforts to ensure reconstruction efficiency. Reconstructing a complete Waymo segment with three front-facing cameras takes 1-1.5 hours, depending on the scene's dynamic complexity. The GPU memory usage is 6GB when sensor data is cached on the CPU and 15GB when cached on the GPU. Tests are performed on a single 4090 GPU.
>
> > Q4. Why does it seem that the reconstruction quality of NuPlan is significantly worse than that of other datasets?
>
> The lower quality on NuPlan is due to severe image distortions of NuPlan’s raw data. Unlike NeRF-based (ray-based rendering) methods, which can account for distortion parameters to correct ray directions, the whole-image rasterization approach of 3DGS does not natively support such undistortion corrections. To the best of our knowledge, no GS-based method currently incorporates distortion parameters for whole-image rasterization.
>
> Fortunately, recent works like 3D Gaussian Ray-Tracing [a] adapt Gaussian to ray-based rendering, making distortion-aware rendering possible. We will integrate this into our framework as a future work, to enhance distortion handling across datasets and improve the system's overall robustness.
>
> [a] 3D Gaussian Ray Tracing: Fast Tracing of Particle Scenes. SIGGRAPH2024

---

> > ### Comment · Reviewer_h6SA · 2024-11-23
> >
> > Thank you to the authors for their responses, and the additional experiments have addressed my concerns. I also consider the future direction to be highly promising, and indeed, there are currently a number of innovative approaches that integrate video generation. Overall, I will maintain my score, as this work is highly complete, and the provided code will greatly contribute to the development of the community and should be accepted.

---

> > > ### Author Response · Authors · 2024-11-23
> > > **Thank you for your feedback**
> > >
> > > Dear reviewer, thank you so much for your valuable feedback and recognition of our paper! We will greatly appreciate an increased score if you find it becomes better---we strive to achieve the strongest possible submission we could have :-) We are committed to addressing any further concerns!

---

### Official Review · Reviewer_dKk5 · 2024-10-31

**Soundness:** 4
**Presentation:** 4
**Contribution:** 4
**Rating:** 8
**Confidence:** 5

**Summary:**

The paper introduces OmniRe, a novel approach for urban scene reconstruction that focuses on dynamic actors, including vehicles, pedestrians, and cyclists. OmniRe employs a Gaussian Scene Graph-based framework to model both static and dynamic objects. To address the limitations of previous methods in reconstructing non-rigid human models, OmniRe integrates SMPL for in-the-wild human representation, allowing for joint-level control. Extensive evaluations across several driving datasets demonstrate OmniRe's superior performance compared to baseline methods.

**Strengths:**

1. The paper is well-organized and easy to follow.
2. The proposed method for in-the-wild human representation is straightforward yet crucial for driving scene reconstruction.
3. Both quantitative and qualitative experiments effectively support the claims made in the introduction, with OmniRe achieving state-of-the-art results across various experimental settings.

**Weaknesses:**

1. Handling Occlusions and Complex Dynamics: OmniRe addresses in-the-wild challenges, yet the performance might be limited by severe occlusions and overlapping actors in complex urban scenes. Further refinement or integration of advanced occlusion handling techniques could enhance reconstruction fidelity.
2. Performance in Specific Urban Scenes: For specialized scenarios, such as highways (with fast-moving vehicles), nighttime environments, and adverse weather conditions, does OmniRe maintain high reconstruction quality under these challenging conditions?

**Questions:**

Please refer to the Weaknesses.

---

> ### Author Response · Authors · 2024-11-22
> **Response to Reviewer dKk5 (1/2)**
>
> We sincerely thank the reviewer for their time and effort in reviewing our paper. We greatly appreciate the positive feedback and are eager to address the points raised. The reviewer's insights are valuable for improving our work. Below, we provide our responses.
>
> > Q1. Handling Occlusions and Complex Dynamics: OmniRe addresses in-the-wild challenges, yet the performance might be limited by severe occlusions and overlapping actors in complex urban scenes. Further refinement or integration of advanced occlusion handling techniques could enhance reconstruction fidelity.
>
> Reconstructing scenes with extreme dynamic occlusions pose significant challenges. However, **OmniRe is designed to tackle these challenges and performs well even in scenes with extreme occlusions.**  We tested OmniRe, StreetGS, PVG, and DeformGS on three extremely occluded scenes from Waymo (seg112520…, seg104859…, seg152664…), including scenarios with a large crowd of people simultaneously crossing a street. OmniRe demonstrated strong performance:
>
> | Method   | Recon Full PSNR/SSIM | Recon Human PSNR/SSIM | Recon Vehicle PSNR/SSIM | Novel Full PSNR/SSIM | Novel Human PSNR/SSIM | Novel Vehicle PSNR/SSIM |
> | :------- | :------------------- | :-------------------- | :---------------------- | :------------------- | :-------------------- | :---------------------- |
> | omnire   | **31.25**/**0.900**  | **27.50**/**0.809**   | **25.91**/**0.812**     | **28.91**/**0.866**  | **23.78**/**0.675**   | **24.46**/**0.750**     |
> | streetgs | 26.81/0.865          | 19.31/0.516           | 24.79/0.787             | 25.67/0.841          | 18.53/0.459           | 23.40/0.723             |
> | pvg      | 29.11/0.865          | 24.56/0.686           | 21.29/0.644             | 27.21/0.830          | 22.24/0.555           | 19.63/0.528             |
> | deformgs | 26.69/0.861          | 19.48/0.504           | 17.49/0.455             | 25.66/0.839          | 18.71/0.448           | 17.12/0.414             |
>
> OmniRe achieved a +2.14 PSNR improvement on scene reconstruction and a +2.96 PSNR improvement on human reconstruction. **We provide additional videos in our [anonymous page](https://anonymousi079j.github.io/omnire_id4816/) (Section: OmniRe on Challenging Scenes).** We will greatly appreciate it if reviewers get a chance to check these additional results of highly challenging scenes.

---

> ### Author Response · Authors · 2024-11-22
> **Response to Reviewer dKk5 (2/2)**
>
> > Q2. Performance in Specific Urban Scenes: For specialized scenarios, such as highways (with fast-moving vehicles), nighttime environments, and adverse weather conditions, does OmniRe maintain high reconstruction quality under these challenging conditions?
>
> The answer is yes-**OmniRe maintains high reconstruction quality under these challenging conditions.** We tested OmniRe, StreetGS, PVG, and DeformGS on various specialized scenes, with detailed results provided below **(videos available in the 'OmniRe on Challenging Scenes' of the [anonymous page](https://anonymousi079j.github.io/omnire_id4816/))**.
>
> a. **Nighttime environments:** We tested three nighttime scenes (seg129008…, seg102261…, seg128560…) from Waymo and observed that OmniRe outperformed the compared methods.
>
> | Method   | Recon Full PSNR/SSIM | Recon Human PSNR/SSIM | Recon Vehicle PSNR/SSIM | Novel Full PSNR/SSIM | Novel Human PSNR/SSIM | Novel Vehicle PSNR/SSIM |
> | :------- | :------------------- | :-------------------- | :---------------------- | :------------------- | :-------------------- | :---------------------- |
> | omnire   | **31.14**/**0.778**  | **31.06**/**0.790**   | **25.87**/**0.774**     | **29.79**/**0.744**  | **25.71**/**0.689**   | 24.29/0.724             |
> | streetgs | 30.60/0.775          | 27.54/0.667           | 25.68/0.772             | 29.38/0.741          | 21.88/0.523           | **24.61**/**0.729**     |
> | pvg      | 30.55/0.768          | 30.74/0.777           | 22.77/0.713             | 29.32/0.740          | 25.22/0.634           | 20.90/0.611             |
> | deformgs | 30.06/0.767          | 27.40/0.665           | 20.62/0.612             | 28.88/0.740          | 22.30/0.530           | 19.20/0.535             |
>
> b. **Adverse weather conditions:** We tested seven scenes under varying weather conditions: 1) rainy: seg113555…, seg109277…, seg141339…; 2) foggy: seg161022…, seg172163…; 3) cloudy: seg144275…, seg157956…. OmniRe handled these challenging scenarios robustly, maintaining high reconstruction fidelity across all weather conditions.
>
> | Method   | Recon Full PSNR/SSIM | Recon Human PSNR/SSIM | Recon Vehicle PSNR/SSIM | Novel Full PSNR/SSIM | Novel Human PSNR/SSIM | Novel Vehicle PSNR/SSIM |
> | :------- | :------------------- | :-------------------- | :---------------------- | :------------------- | :-------------------- | :---------------------- |
> | omnire   | **34.00**/**0.935**  | **30.12**/**0.857**   | **32.55**/**0.919**     | **32.58**/**0.920**  | **26.75**/**0.763**   | **29.82**/**0.844**     |
> | streetgs | 33.49/0.933          | 18.26/0.424           | 32.28/0.916             | 32.05/0.918          | 18.56/0.434           | 29.60/0.841             |
> | pvg      | 32.75/0.927          | 27.75/0.785           | 27.20/0.795             | 31.12/0.907          | 25.02/0.656           | 24.44/0.677             |
> | deformgs | 32.92/0.933          | 20.06/0.497           | 23.12/0.658             | 31.43/0.916          | 19.65/0.478           | 22.06/0.598             |
>
> c. **Highway scenes:** We tested three highway scenes from the Waymo dataset (seg109239…, seg113924…, seg118396). Highways typically lack non-rigid classes (e.g., humans), making human reconstruction metrics inapplicable. As non-rigid modeling was not required in these scenes, OmniRe and StreetGS demonstrated similar performance.
>
> | Method   | Recon Full PSNR/SSIM | Recon Human PSNR/SSIM | Recon Vehicle PSNR/SSIM | Novel Full PSNR/SSIM | Novel Human PSNR/SSIM | Novel Vehicle PSNR/SSIM |
> | :------- | :------------------- | :-------------------- | :---------------------- | :------------------- | :-------------------- | :---------------------- |
> | omnire   | 30.85/**0.902**      | -/-                   | **29.51**/0.882         | 28.01/**0.864**      | -/-                   | 24.52/0.760             |
> | streetgs | **30.89**/**0.902**  | -/-                   | 29.45/**0.883**         | **28.05**/0.863      | -/-                   | **24.88**/**0.766**     |
> | pvg      | 28.28/0.861          | -/-                   | 21.14/0.617             | 25.79/0.825          | -/-                   | 17.65/0.470             |
> | deformgs | 26.95/0.855          | -/-                   | 17.49/0.482             | 25.56/0.837          | -/-                   | 16.07/0.407             |

---

> > ### Comment · Reviewer_dKk5 · 2024-11-24
> > **Response to Rebuttal**
> >
> > Thanks for your reply! I will keep my rating.

---

> > > ### Author Response · Authors · 2024-11-25
> > > **Rebuttal follow up**
> > >
> > > Dear reviewer,
> > >
> > > Thank you for your follow-up and for taking the time to review our updates. We appreciate your efforts! We hope that our additional results on the challenging scenes, along with the provided visualizations, have addressed your concerns about OmniRe's ability to handle challenging scenes. These results aim to provide a more comprehensive insight and strengthen our paper.
> > >
> > > We will greatly appreciate an increased score if you find it becomes better, we strive to achieve the strongest possible submission we could have. We are committed to addressing any further concerns!

---

### Official Review · Reviewer_WVni · 2024-11-04

**Soundness:** 3
**Presentation:** 3
**Contribution:** 3
**Rating:** 6
**Confidence:** 3

**Summary:**

OmniRe is a framework designed to create high-fidelity digital twins of dynamic urban scenes for simulations, particularly for applications in autonomous driving. OmniRe goes beyond vehicle modeling to support diverse dynamic actors like pedestrians and cyclists, enabling complex simulations that reflect real-world scenarios. It utilizes Gaussian Scene Graphs with multiple representations, allowing detailed and editable scene reconstructions with both rigid (e.g., vehicles) and non-rigid (e.g., pedestrians) actors.

**Strengths:**

Comprehensive Dynamic Modeling: OmniRe can handle various actors in urban settings, unlike most previous methods that focus mainly on vehicles.

Scene Graphs and Gaussian Splatting: The system uses 3D Gaussian splatting for detailed scene and object rendering, including control over each object.

Human Behavior Simulation: Through SMPL modeling, OmniRe accurately reconstructs human motions, even in cluttered environments, enabling simulations of interactions between pedestrians and vehicles.

State-of-the-Art Performance: Extensive testing on datasets like Waymo and others show OmniRe significantly outperforms existing methods in terms of visual fidelity and reconstruction accuracy.

**Weaknesses:**

There are several limitations in OmniRe approach, which are correctly identified in the paper too.

Lighting Effects: OmniRe doesn’t model lighting variations explicitly. This can lead to visual inconsistencies when combining scene elements with differing lighting conditions, which may reduce realism in certain simulations. Addressing this would require additional modeling of lighting dynamics.

Novel View Synthesis Limitations: OmniRe’s per-scene optimization approach struggles to generate satisfactory results when the camera view deviates significantly from the training trajectories. This could be a limitation for scenarios requiring a wide range of viewing angles, such as free navigation through the reconstructed scenes. The authors suggest incorporating data-driven priors or generative models as future work to address this.

Computational Complexity: While the method achieves high-quality reconstructions, the complexity of the Gaussian Scene Graph and the joint optimization of multiple parameters (pose, appearance, etc.) require substantial computational resources. Training time per scene, though optimized for an RTX 4090 GPU, could still pose scalability issues for large datasets or continuous real-time simulation needs.

Challenges with Real-Time Adaptability: The method’s reliance on SMPL modeling for human actors and per-node deformation fields, though effective, might introduce delays in real-time applications, particularly if scenes are highly dynamic or involve many non-rigid actors.

**Questions:**

I wonder for items like causality and new synthesis if an approach more configurable could take place now that they have separated the pedestrians from the road.

Thinking of something like this
Wang, Cheng Yao, et al. "CityLifeSim: A High-Fidelity Pedestrian and Vehicle Simulation with Complex Behaviors." 2022 IEEE 2nd International Conference on Intelligent Reality (ICIR). IEEE, 2022.

Where the data is later attached to an engine like Carla or airsim

---

> ### Author Response · Authors · 2024-11-22
> **Response to Reviewer WVni (1/2)**
>
> We sincerely thank the reviewer for the time and effort reviewing. **We are encouraged by the reviewer's recognition of the value our method brings to dynamic scene modeling, human-level reconstruction and simulation, and SOTA performance.** Our replies are stated below.
>
> > There are several limitations in OmniRe approach, which are correctly identified in the paper too.
> >
> > Q1. Lighting Effects: OmniRe doesn’t model lighting variations explicitly. This can lead to visual inconsistencies when combining scene elements with differing lighting conditions, which may reduce realism in certain simulations. Addressing this would require additional modeling of lighting dynamics.
>
> Thanks for pointing out its importance and for recognizing that we have acknowledged it in the limitation section. Modeling lighting effects to enhance simulation realism is crucial for achieving more convincing and harmonious results. Solving it is non-trivial and requires dedicated efforts. For example, LightSim [a] is a standalone paper that focuses specifically on this issue.
>
> While this is slightly beyond the scope of our current work, we would like to discuss a few potential directions for light effect modeling based on our current framework: 1) incorporate BRDF model to learn reflectance, material properties, and utilize environmental maps for illumination modeling; 2) post-processing rendered images with harmonization techniques, such as rendering feature maps and training a network to convert these maps into photorealistic RGB images. 3) using video diffusion models to refine the outputs, e.g. localized editing with diffusion models [b].
>
> [a] LightSim: Neural lighting simulation for urban scenes. NeurIPS2023.
> [b] Stable Video Diffusion: Scaling Latent Video Diffusion Models to Large Datasets.
>
> > Q2. Novel View Synthesis Limitations: OmniRe’s per-scene optimization approach struggles to generate satisfactory results when the camera view deviates significantly from the training trajectories. This could be a limitation for scenarios requiring a wide range of viewing angles, such as free navigation through the reconstructed scenes. The authors suggest incorporating data-driven priors or generative models as future work to address this.
>
> Driving scenes, which typically follow a forward-moving trajectory, often involve sparser observations compared to object-centric scenes. For driving simulation, simulated viewpoints are typically constrained to those on the road, with acceptable deviation. As a result, novel view rendering maintains high quality, as shown in the top-right novel-view navigation demo on the anonymous page.
>
> While the current novel view quality meets the requirements for typical driving simulations (e.g., lane shifts or various vehicle heights), limitations arise for free navigation with viewpoints far outside the training space. For future work, we believe methods using generative models like [c] could enhance reconstruction by inferring missing details and completing unseen regions.
>
> [c] Streetscapes: Large-scale Consistent Street View Generation Using Autoregressive Video Diffusion. SIGGRAPH2024

---

> ### Author Response · Authors · 2024-11-22
> **Response to Reviewer WVni (2/2)**
>
> > Q3. Computational Complexity
> >
> > & Q4. Challenges with Real-Time Adaptability
>
> **Training Efficiency:** Although our system integrates multiple Gaussian representations for full category reconstruction and jointly optimizes the appearance and poses of all objects (dozens or even over a hundred) along with the background, **we made substantial engineering efforts for the whole system to ensure its efficiency and robustness,** including:
>
> - Parallelizing the deformation and transformation of each object's Gaussians using category-specific Gaussian containers. These containers handle all objects' Gaussians and compute deformations in parallel. For example, the vehicle class container processes all vehicles’ Gaussians simultaneously.
> - Sharing the weights and gradients of the deformation network for non-rigid deformable nodes and using a shared network with multiple instance embeddings to distinguish them.
>
> **Rendering Efficiency:**  The rendering of OmniRe operates in **real-time** (>24 fps), meeting most simulation requirements. To address the concern regarding its real-time adaptability, we conducted tests under various settings with increasing dynamic complexity:
>
> - Background only: 170 fps
>
> - Background + Vehicle (Rigid Nodes): 108 fps
>
> - Background + Vehicle + SMPL-GS (Non-Rigid Nodes): 57 fps
>
> - Background + Vehicle + Deformable-GS (Non-Rigid Nodes): 68 fps
>
> - Background + Vehicle + Deformable-GS + SMPL-GS: 45 fps
>
>   (Experiments conducted on a single 4090 GPU)
>
> > Q5. I wonder for items like causality and new synthesis if an approach more configurable could take place now that they have separated the pedestrians from the road. Thinking of something like CityLifeSim [a], where the data is later attached to an engine like Carla or airsim.
>
> The idea behind CityLifeSim [a] is highly inspiring and crucial for an advancing simulation system, especially in aspects like modeling causality and designing configurable actor behaviors. We will explore ways to apply its ideas to our 3DGS-based framework.
>
> The waypoint-based POI (points of interest) for actor interaction rule design presented in CityLifeSim [a] is adaptable to our framework with some additional work, such as labeling valid activity areas for each object. However, the primary challenge lies in the animation of humans, since we use a different modeling approach (SMPL), animating humans in our framework requires additional attention to motion naturalness. Ensuring realistic motion through effective SMPL parameter control is a non-trivial task. Techniques like text-driven motion [b] models could enhance motion naturalness and contribute to the design of more robust interaction rules.
>
> Thank you again for highlighting this important work. We have included a discussion of CityLifeSim in the revised manuscript **line 511-513**.
>
> [a] CityLifeSim: A High-Fidelity Pedestrian and Vehicle Simulation with Complex Behaviors. ICIR2022
> [b] Generating Diverse and Natural 3D Human Motions from Text. CVPR2022

---

> ### Author Response · Authors · 2024-11-25
> **Rebuttal follow up**
>
> Dear reviewer,
>
> We greatly appreciate the time and effort you have invested in reviewing our work! We hope the additional experiments on OmniRe’s real-time rendering and simulation capabilities address concerns regarding computational complexity and real-time performance. This work involved substantial engineering efforts to ensure the system's efficiency and robustness.
>
> We are also greatly inspired by your suggestion regarding CityLifeSim to further explore the possible extension of our work. Based on our current reconstruction framework, we will incorporate elements like configurable actor behaviors and interaction rules to develop a more comprehensive simulator in future work.
>
> We understand how busy you must be, especially as we approach the end of the discussion period. We will greatly appreciate an increased score if we successfully address your concerns. Otherwise, we are committed to provide additional results to answer any further questions.

---

> ### Author Response · Authors · 2024-11-29
> **Rebuttal follow up**
>
> Dear Reviewer,
>
> Thank you again for helping us improve the paper. As the rebuttal period is coming to an end, we would like to check if our responses have addressed your concerns. If so, we would greatly appreciate it if you could consider raising the score accordingly. Otherwise, we would be grateful for any additional comments or questions, and we would be happy to include further experiments and results. Thanks!

---

### Author Response · Authors · 2024-11-22
**General Response**

We sincerely thank the reviewers for their time and effort in reviewing our manuscript, as well as for their insightful feedback and recognition of the strengths of our work:

- **Comprehensive Dynamic Modeling:** Our work presents a holistic framework that unifies the reconstruction of static backgrounds, driving vehicles, and non-rigidly moving dynamic actors. (WVni, h6SA)
- **Fine-Grained Reconstruction and Behavior Simulation:** We address the challenges of modeling humans and other dynamic actors from driving logs, even in cluttered environments. (WVni, dKk5)
- **Extensive Results and SOTA Performance:** Our method is demonstrated on six popular driving datasets, achieving SOTA performance in scene reconstruction and novel view synthesis. (WVni, dKk5, h6SA)
- **Dedicated Engineering Effort and Community Value:** We are committed to providing a complete codebase and holistic framework, which holds significant value for advancing autonomous driving simulation. (h6SA)

We have revised the main paper and supplementary material according to your comments, with all revisions highlighted in **red**.

---

**OmniRe on Highly Challenging Scenes:** We performed additional experiments on various types of challenging dynamic scenes: 1) Super Crowded Scenes, 2) Nighttime Scenes, 3) Adverse Weather Conditions, 4) High-Speed Scenes. OmniRe performs well across these scenarios, demonstrating strong performance (quantitative results in the revised manuscript, page 18). **These results highlight the overall robustness and generalizability of OmniRe to diverse driving scenes.**

We updated the visualizations in section **"OmniRe on Challenging Scenes"** on our [anonymous page](https://anonymousi079j.github.io/omnire_id4816/)  for better understanding. We will greatly appreciate it if reviewers get a chance to check these additional results of highly challenging scenes.

---

### Meta-Review · Area_Chair_uJ3v · 2024-12-23

**Metareview:**

The paper presents an approach to simulate the appearance of dynamic road scenes using sene graphs on 3D Gaussian splats. Although the technical methods are not novel, the paper combines multiple tools into a pipeline that produces good results in demonstrated applications. The authors are encouraged to include rebuttal discussions in the final version and the paper is recommended for acceptance at ICLR.

**Additional Comments On Reviewer Discussion:**

While WVni correctly points out limitations in the proposed method to correctly handle lighting variations or large view variations, the AC agrees with the rebuttal that the scope of the paper is already sufficiently broad. Additional experiments in response to dKk5 are recommended to the added to the paper. The rebuttal is acknowledged by h6SA as convincing, especially as the code is mentioned as to be released by the authors.

---

### Decision · Program_Chairs · 2025-01-22

Accept (Spotlight)